# LOBBEN-TM: A BENCHMARK STUDY OF LIMIT ORDER BOOK PREDICTION WITH TEMPORAL MODELING

## ABSTRACT

We introduce LOBBen-TM, **L**imit **O**rder **B**ook (LOB) **Ben**chmark with **T**emporal **M**odeling, for deep learning on open-sourced LOB data that unifies evaluations across tasks, features, and assets. Our work makes four major contributions: (i) On the Mid-Price Trend Prediction (MPTP) task, we assess state-of-the-art LOB models with a standardized full LOB feature set with time-sensitive features on two assets, equities (FI-2010) and cryptocurrency (Bitcoin), to probe cross-asset generalization under a common protocol. We further benchmark a common LOB feature taxonomy (basic, time-insensitive, time-sensitive) and conduct an ablation on FI-2010. (ii) We extend the study to Mid-Price Return Forecasting (MPRF), jointly evaluating LOB-specific architectures and top-tier general time-series prediction models on FI-2010 with MSE, R2, and Pearson correlation. (iii) To enhance multivariate time series prediction models on LOB returns, we propose a lightweight Cross-Variate Mixing Layer (CVML) that plugs into existing models. Empirically, results show that the standardized full feature set yields robust MPTP performance across FI-2010 and Bitcoin while revealing asset-dependent ranking shifts. Besides, time-sensitive features provide sizable improvements on FI-2010, underscoring the importance of temporal signal modeling. Last but not least, our proposed CVML architecture substantially boosts general time series prediction models on MPRF, narrowing the gap to LOB models and advancing return forecasting on LOB data.

## 1 INTRODUCTION

The Limit Order Book (LOB) serves as the order-matching engine for all exchanges, providing the most granular market time-series data for analysis (Weber, 1999; Ntakaris et al., 2018). As a universal data format across markets and assets (e.g., equities and cryptocurrencies), LOB contains high-resolution macro/micro-structural information crucial for asset price prediction (Chan et al., 2005; Harris & Panchapagesan, 2005; Large, 2007; Avellaneda & Stoikov, 2008; Roşu, 2009; Eisler et al., 2012).

Two primary research tracks in LOB analysis are **M**id-**P**rice **T**rend **P**rediction (**MPTP**) and **M**id-**P**rice **R**eturn **F**orecasting (**MPRF**). Inspired by the success of deep learning in NLP (Vaswani et al., 2023) and computer vision (He et al., 2016), researchers have proposed deep learning models for LOB analysis and general time series predictions (Zeng et al., 2023; Liu et al., 2023; Nie et al., 2022; Wang et al., 2024).

Yet comprehensive benchmarks remain scarce. For MPTP, Prata et al. (2024) provides a valuable study, but it focuses on equities and does not cover other asset classes such as cryptocurrencies. For MPRF, sitting at the intersection of LOB modeling and time-series forecasting, there is still no benchmark jointly evaluating LOB-specific models and state-of-the-art (SOTA) general time-series predictors on LOB data.

This work addresses these gaps in a staged manner. We conduct a comprehensive study on both MPTP and MPRF tasks. First, on the MPTP task we benchmark SOTA LOB models using a standardized *full LOB feature set* across two datasets of different assets: the open FI-2010 equities dataset and a cryptocurrency LOB dataset (Bitcoin). This cross-asset evaluation probes the *generalization* of a common feature pipeline and LOB models under a unified protocol with multiple prediction horizons. Second, still on MPTP, we perform a controlled ablation on FI-2010 to quantify the

incremental value of *time-sensitive features*, a subset within the common feature taxonomy (basic LOB, time-insensitive, time-sensitive). The substantial gains from time-sensitive features emphasize the importance of explicitly modeling temporal signal structure within LOBs.

Motivated by this observation and recognizing that LOB is inherently a time-series signal, we next investigate whether top-tier general time-series prediction models transfer effectively to LOB *returns*. We therefore extend the benchmark to the MPRF task and jointly evaluate LOB-specific architectures alongside leading time-series backbones on FI-2010, using Mean-Squared Error (MSE), Coefficient of Determination ($R^2$), and Pearson Correlation (Corr) as evaluation metrics. To further strengthen general time-series models on multivariate LOB inputs, we propose a lightweight **C**ross-**V**ariate **M**ixing **L**ayer (**CVML**) that can be attached to existing time series prediction models.

Our experiments yield three takeaways. (i) On MPTP, the standardized full feature set delivers robust performance across assets (FI-2010 and Bitcoin), while performance levels and model rankings can still vary by asset, underscoring both the utility of a shared feature pipeline and the heterogeneity of LOB dynamics across markets. (ii) Time-sensitive features provide consistent, sizable gains on FI-2010, highlighting the centrality of temporal modeling in LOB prediction. (iii) On MPRF, LOB-specific inductive biases remain advantageous while general-purpose time series prediction models show extremely limited forecasting power. However, empirically, CVML substantially boosts the competitiveness of general time-series prediction models, narrowing the gap on LOB returns.

**Contributions.** We summarize our contributions as follows:

1. **Cross-asset MPTP benchmark.** We evaluate state-of-the-art LOB models with a standardized *full* feature set on two assets, equities (FI-2010) and cryptocurrency (Bitcoin), to assess feature-pipeline and model generalization across markets.
2. **Feature taxonomy and ablation.** We benchmark the common LOB feature set (basic, time-insensitive, time-sensitive) and isolate the incremental benefit of *time-sensitive* features on FI-2010 for MPTP.
3. **First joint MPRF benchmark of LOB and TS models.** We jointly evaluate LOB-specific architectures and top-tier general time-series predictors on LOB MPRF, reporting MSE/$R^2$/Corr under a unified protocol.
4. **CVML for cross-variate signal mixing.** We propose a plug-in Cross-Variate Mixing Layer that consistently improves general time-series backbones on LOB MPRF.

These contributions advance LOB modeling across different assets and tasks while providing a new tool to enhance time series model performance on LOB data.

We organize the rest of the paper as follows. Section 2 discusses background information on the MPTP and MPRF tasks. Section 3 details our benchmark studies and our proposed CVML architecture. Section 4 includes the conclusion.

## 2    PROBLEM DEFINITION

We model the LOB as a time series $\mathbb{L} \in \mathbb{R}^{4 \times L \times T}$, where $\mathbb{L}(t) \in \mathbb{R}^{4 \times L}$ represents the LOB at time step $t \in [0, T]$. Specifically, $\mathbb{L}(t) = \{P_i^{\text{bid}}(t), Q_i^{\text{bid}}(t), P_i^{\text{ask}}(t), Q_i^{\text{ask}}(t)\}_{i \in [0, L]}$, with $T$ observed time steps and $L$ levels on each side of the order book. $P^{\text{bid/ask}}i(t)$ and $Q_i^{\text{bid/ask}}(t)$ denote the price and quantity at level $i$ at time $t$, respectively. Levels are ordered by price, representing order priority in matching. The best bid price is the highest bid, while the best ask is the lowest ask. The mid-price, $\text{mp}(t)$, is the average of these best prices. Execution of more bid (ask) orders decreases (increases) the mid-price.

**Mid-Price Return Forecasting (MPRF).** To address mid-price volatility and non-stationarity, we model the problem as forecasting the mid-price *return*. At time $t$, our target is:

$$\text{target}_h(t) = \text{mp}(t + h)/\text{mp}(t) - 1,$$

where $h$ is the forecasting horizon.

**Mid-Price Trend Prediction (MPTP).** An alternative approach models mid-price prediction as a classification problem, categorizing trends into three classes: **U** (upward), **D** (downward), and **S** (stable). Following (Ntakaris et al., 2018), we generate labels from raw LOB data by comparing the

current mid-price to the average future mid-price:

$$
\begin{cases}
\mathbf{U}, & \text{if} \quad \text{avg\_mp}(k,t) > \text{mp}(t) \times (1 + \alpha) \\
\mathbf{D}, & \text{if} \quad \text{avg\_mp}(k,t) < \text{mp}(t) \times (1 - \alpha) \\
\mathbf{S}, & \text{if} \quad \text{Otherwise,}
\end{cases}
$$

where $\text{avg\_mp}(k,t) = (\Sigma_{i=1}^{k}\text{mp}(t+i))/k$. Using $\alpha = 0.002\%$ yields approximately equal distribution (33%) for each label.

## 3 EXPERIMENTS

We conduct all experiments using 3 seeds to minimize the effect of random initialization.

**Datasets**: **1) FI-2010** (Ntakaris et al., 2018)[1]: This Limit Order Book (LOB) dataset includes 10 trading days of data from five Finnish companies on the NASDAQ Nordic stock market. It was designed to evaluate machine learning models' performance on stock price trend prediction. **2) Bitcoin**: To evaluate the generalization ability of the LOB features, we extend the dataset to a different asset by including a public Bitcoin LOB dataset [2]. It covers 12 days of Bitcoin limit order book of a frequency of 250 milliseconds.

**Evaluation Metrics:** For MPTP, we use the F1 scores. For MPRF, we use Mean Squared Error (MSE), Pearson Correlation Coefficient (Corr), and Coefficient of Determination ($R^2$). MSE quantifies the average squared difference between predicted and actual returns, providing a measure of prediction accuracy. The Pearson Correlation Coefficient assesses the linear relationship between predicted and actual returns, indicating the direction and strength of their association. Lastly, $R^2$ represents the proportion of variance in the target variable explained by our model. The implementation of MSE and $R^2$ are from Scikit-learn (Pedregosa et al., 2011). The implementation of Pearson Correlation Coefficient is from SciPy (Virtanen et al., 2020).

**Hyperparameter Search.** For MPTP, we use the hyperparameters from the original paper of the models. For MPRF, we perform a grid search. Details are in Appendix D.

Table 1: **Input lookback size and number of features for MPTP Models.** (Tsantekidis et al., 2017b;b;a; Tran et al., 2018; Zhang et al., 2019; Passalis et al., 2019; Tsantekidis et al., 2020; Wallbridge, 2020; Passalis et al., 2020; Zhang & Zohren, 2021; Guo & Chen, 2022) The number of features is formatted as [basic]/[basic + time-insensitive]/[basic + time-insensitive + time-sensitive]

|  | MLP | LSTM | CNN1 | CTABL | DEEPLOB | DAIN | CNNLSTM | CNN2 | TRANSLOB | TLONBoF | BINCTABL | DEEPLOBATT | DLA |
|---|---|---|---|---|---|---|---|---|---|---|---|---|---|
| Lookback size | 100 | 100 | 100 | 10 | 100 | 15 | 300 | 300 | 100 | 15 | 10 | 50 | 5 |
| Features (FI-2010) | 40/86/144 | 40/86/144 | 40/86/144 | 40/86/144 | 40/86/144 | 40/86/144 | 40/86/144 | 40/86/144 | 40/86/144 | 40/86/144 | 40/86/144 | 40/86/144 | 40/86/144 |

Table 2: **Input lookback size and number of features for MPRF Models.** (Nie et al., 2022; Zeng et al., 2023; Liu et al., 2023; Wang et al., 2024) The number of features is formatted as [basic]/[basic + time-insensitive]/[basic + time-insensitive + time-sensitive]

|  | MLP | LSTM | CNN1 | BINCTABL | DAIN | TRANSLOB | PatchTST | DLinear | iTransformer | TimeMixer |
|---|---|---|---|---|---|---|---|---|---|---|
| Lookback size | 100 | 100 | 100 | 10 | 15 | 100 | 100 | 100 | 100 | 100 |
| Features (FI-2010) | 41/86/144 | 41/86/144 | 41/86/144 | 41/86/144 | 41/86/144 | 41/87/145 | 41/87/145 | 41/87/145 | 41/87/145 | 41/87/145 |

### 3.1 MODELS

Our benchmark includes models for two tasks: MPTP and MPRF. For MPTP, we select 13 state-of-the-art models. The input and output of each mid-price trend prediction model follow the same protocol in their original paper. For MPRF, we choose 10 models with two goals in mind: benchmarking diverse neural architectures (MLP, CNN, LSTM, and Transformer) and evaluating the importance of domain-specific inductive bias for LOB data. We include some models from the MPTP list that have LOB-specific adaptations, as well as high-performing general-purpose time series forecasters. This mix allows us to compare specialized LOB models against successful general-purpose forecasters. For time series forecasting models including PatchTST, DLinear, iTransformer and TimeMixer, the

---

[1]License: Creative Commons Attribution 4.0 International (CC BY 4.0)

[2]https://tinyurl.com/mv68vheb

Table 3: **Mid-price Trend Prediction F1 Scores (Mean&Standard Deviation) on Basic LOB data + time-insensitive features + time-sensitive features**. We provide the F1 scores on mid-price trend predictions across horizons {1,2,3,5,10} for the FI-2010 and Bitcoin datasets. The model performance ranking is not consistent between two datasets, indicating that models' prediction power for one asset is not automatically transferable to another asset.

| | FI-2010 | | | | | | Bitcoin | | | | | |
|---|---|---|---|---|---|---|---|---|---|---|---|---|
| Model | K=1 | K=2 | K=3 | K=5 | K=10 | avg | K=1 | K=2 | K=3 | K=5 | K=10 | avg |
| MLP | 54.0 (3.7) | 52.6 (2.0) | 55.6 (0.3) | 55.0 (0.4) | 52.3 (2.8) | 53.9 | 92.4 (0.2) | 93.2 (0.1) | 93.8 (0.1) | 94.3 (0.1) | 95.3 (0.0) | 93.8 |
| LSTM | 76.3 (0.3) | 75.4 (0.1) | 76.9 (0.9) | 75.6 (1.1) | 62.9 (1.3) | 73.4 | 86.2 (2.8) | 94.7 (0.2) | 95.2 (0.5) | 96.3 (0.1) | 97.2 (0.1) | 94.0 |
| CNN1 | 73.6 (0.3) | 72.0 (0.8) | 75.3 (0.5) | 78.4 (0.8) | 78.9 (1.1) | 75.6 | 94.5 (0.7) | 96.3 (0.2) | 96.9 (0.2) | 96.9 (0.1) | 97.7 (0.1) | 96.5 |
| CTABL | 76.6 (0.2) | 72.6 (0.4) | 77.9 (0.2) | 82.1 (0.5) | 83.3 (0.5) | 78.5 | 53.9 (0.2) | 64.3 (0.0) | 72.1 (0.2) | 80.9 (0.2) | 92.3 (0.2) | 72.7 |
| DEEPLOB | **81.2 (0.3)** | 82.4 (0.7) | 86.1 (0.5) | 88.1 (0.2) | 88.7 (0.2) | 85.3 | **97.9 (0.1)** | **98.4 (0.0)** | 98.5 (0.1) | 98.1 (0.0) | 98.3 (0.0) | 98.3 |
| DAIN | 80.8 (0.1) | 79.7 (0.1) | 85.8 (0.1) | 90.0 (0.0) | 93.2 (0.0) | 85.9 | 34.7 (0.3) | 53.3 (0.4) | 66.9 (0.6) | 79.8 (0.1) | 87.1 (0.1) | 64.4 |
| CNNLSTM | 74.1 (0.4) | 68.1 (0.3) | 73.7 (0.7) | 77.3 (1.1) | 77.8 (0.9) | 74.2 | 97.2 (0.1) | 97.7 (0.2) | 97.9 (0.1) | 97.7 (0.1) | 98.1 (0.0) | 97.7 |
| CNN2 | 73.1 (0.2) | 67.5 (1.0) | 70.8 (4.2) | 74.4 (3.4) | 65.7 (1.2) | 70.3 | 97.1 (0.1) | 98.1 (0.0) | 98.2 (0.1) | 97.9 (0.1) | 98.1 (0.0) | 97.9 |
| TRANSLOB | 74.5 (1.1) | 70.5 (0.3) | 75.2 (0.3) | 78.4 (0.2) | 68.2 (3.2) | 73.4 | 95.9 (0.8) | 98.2 (0.2) | 98.3 (0.2) | 98.0 (0.2) | 98.4 (0.1) | 97.8 |
| TLONBoF | 61.3 (6.1) | 66.9 (0.3) | 72.6 (0.7) | 75.2 (4.0) | 71.2 (2.7) | 69.4 | 56.5 (1.0) | 70.0 (0.9) | 78.1 (0.7) | 88.2 (0.3) | 94.8 (0.1) | 77.5 |
| BINCTABL | 80.3 (0.1) | **83.4 (0.3)** | **87.9 (0.2)** | **91.3 (0.1)** | **93.2 (0.5)** | 87.2 | 50.5 (0.4) | 60.3 (0.5) | 65.1 (1.7) | 73.1 (0.2) | 90.4 (0.4) | 67.9 |
| DEEPLOBATT | 77.9 (0.0) | 78.5 (0.0) | 82.5 (0.0) | 82.9 (0.0) | 82.1 (0.0) | 80.8 | 97.6 (0.2) | 98.1 (0.5) | **98.7 (0.2)** | **98.4 (0.1)** | **98.5 (0.1)** | 98.3 |
| DLA | 71.8 (0.0) | 71.1 (0.0) | 76.4 (0.0) | 85.0 (0.0) | 59.3 (0.0) | 72.7 | 57.1 (2.0) | 69.2 (0.3) | 74.4 (0.4) | 81.0 (0.2) | 89.0 (0.2) | 74.2 |
| avg | 73.5 | 72.4 | 76.7 | 79.5 | 75.1 | 75.4 | 77.8 | 84.0 | 87.2 | 90.8 | 95.0 | 87.0 |

LOB data as well as the history mid-price return are input as a multivariate time series. The output for all MPRF models is a scaler representing the return of horizon $h$. Each input is $\mathbb{R}^{T \times 4L}$, output is $\mathbb{R}$. Detailed model and input information is in Appendix C. Table 1 and Table 2 include the input lookback size and feature dimension for each model.

## 3.2 MID-PRICE TREND PREDICTION RESULTS

Table 3 reveals inconsistencies in the performance ranking of the models between the stock FI-2010 and Bitcoin datasets. The ordering on FI-2010 does not carry over to Bitcoin, suggesting market-microstructure–dependent inductive biases.

To investigate the predictive power of different feature types (basic, time-insensitive, time-sensitive), we evaluate the models on two feature subsets: one with only basic features, and another with basic and time-insensitive features. Figure 1 illustrates the gains in prediction performance based on average F1 scores across 5 horizons for each feature set (full results are available in Appendix F). We define feature set gains as the difference between its average F1 scores and those of the basic features. Consequently, basic features have zero gain, serving as the baseline. Positive gains for time-insensitive and time-sensitive features indicate additional predictive power beyond raw LOB readings. Figure 1 demonstrates that both time-insensitive and time-sensitive features contribute positive gains on top of basic LOB features for the FI-2010 dataset. The positive gain from time-sensitive features further indicates opportunities from general-purpose time series forecasting models for the MPRF task on LOB data.

## 3.3 MID-PRICE RETURN FORECASTING RESULTS

Table 4 shows that LOB models with LOB-specific inductive bias significantly outperform general-purpose time series prediction models. This highlights the importance of incorporating LOB-related inductive bias in model design. Additionally, the results indicate that general-purpose models lack the generalization needed for strong forecasting performance on LOB datasets. This performance gap underscores the specialized nature of LOB data and the need for tailored models in financial forecasting.

**Prediction Performance Gap Between LOB Models and Time Series Models.** In the MPRF task, a significant performance gap exists between LOB-specific models and general time series models. Notably, complex Transformer-based models including PatchTST and iTransformer underperform compared to simpler LOB-specific architectures based-on MLPs or LSTMs. This discrepancy suggests that without LOB-aware architectural design, conventional time series models struggle to generate accurate predictions on LOB data due to its low signal-to-noise ratio. This observation indicates that the sophisticated temporal modeling capabilities of state-of-the-art time series models may be impeded by the noisy temporal dynamics and intricate cross-variate correlations inherent in LOB data. Based on this hypothesis, we propose a novel module called Cross-Variate Mixing Layers (CVML) to enhance the signal-to-noise ratio of LOB time series. CVML serves as an add-on

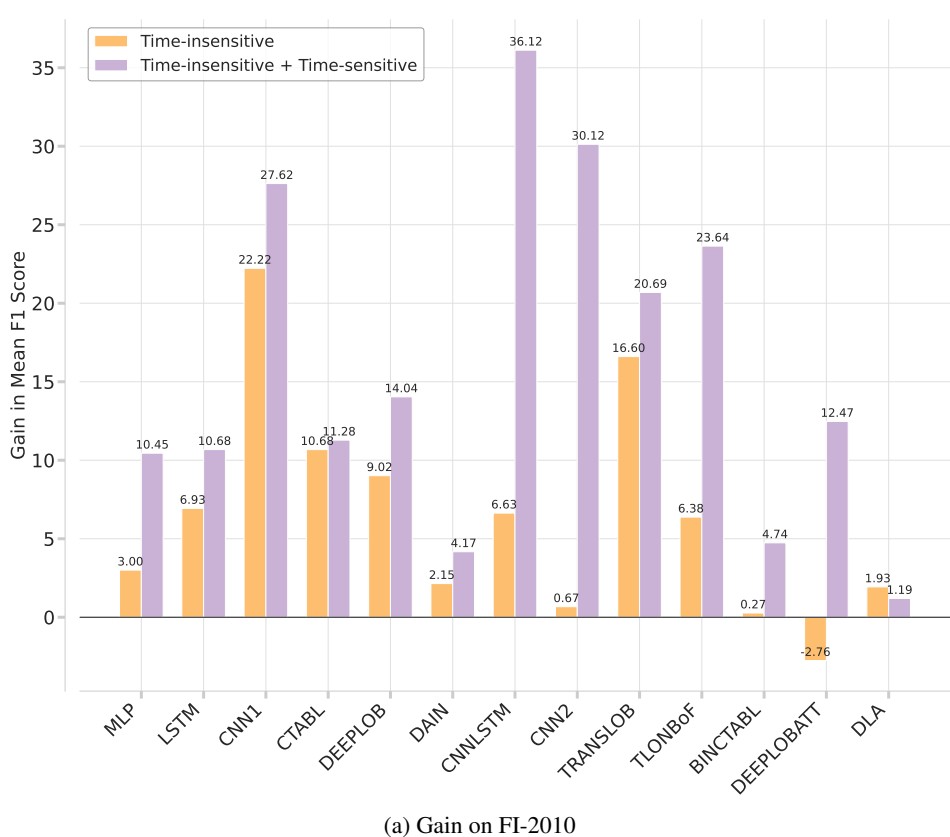

(a) Gain on FI-2010

Figure 1: **Gains in mean F1 scores.** This plot illustrates the incremental prediction power gained from time-insensitive and time-sensitive features on FI-2010 dataset. Yellow bars indicate the improvement in F1 scores when adding time-insensitive features to basic LOB features for most models. Purple bars, compared to yellow, demonstrate the further enhancement in F1 scores when incorporating time-sensitive features alongside basic and time-insensitive features.

layer preceding the time series modeling layers in standard time series prediction models. It accepts raw LOB data, mixes the features (or different variates from a time series perspective), and outputs an intermediate multivariate time series as input for subsequent modeling layers. CVML integrates seamlessly with existing time series models without requiring further modifications and can be trained end-to-end. CVML has five Conv1D layers, each with a kernel size of 2 and $\lceil N/2 \rceil$ output channels, where $N$ is the number of input features. This design leverages convolution kernels to extract cross-variate features and capture correlations. Additionally, we implement increasing dilation in the kernel for each successive layer to enhance temporal signal extraction.

To demonstrate CVML's efficacy, we prepend it to four time series models without altering their core architectures. Results show significant improvements in forecast performance across all metrics (MSE, Corr, and $R^2$), as shown in Table 5. Averaging across four models and five horizons, CVML achieves a 7.4% improvement in MSE, 101.6% in Pearson Correlation (Corr), and 244.9% in $R^2$. This highlights CVML's potential as a powerful add-on for enhancing general time series models on complex LOB data, bridging the gap between specialized LOB models and general-purpose forecasting approaches.

To illustrate CVML's effects, we analyze the standard deviation ($std$) of the time steps in each mid-price return input, then plot the histogram of these $std$ values across all inputs. Figure 3 demonstrates that CVML significantly reduces $std$ values of the LOB time series, indicating its effective smoothing effect. Furthermore, the distribution in Figure 3b more closely resembles a normal distribution. These transformations collectively enhance the time series models' ability to capture temporal signals by presenting them with more structured and less noisy data. This visualization provides evidence of

Table 4: **Mid-price Return Forecasting Results (Mean) on Basic LOB data**. 10 LOB models and time series forecasting models are benchmarked to compare their Mean Square Error (MSE), Pearson correlation (Corr), and Coefficient of Determination ($R^2$) on mid-price return forecasting across 5 horizons {1,2,3,5,10} on the FI-2010 dataset. LOB models perform much better than general-purpose time series models, indicating that it is essential to include LOB-relevant inductive bias into the model design to achieve good forecasting power on LOB datasets. For each horizon, the best model is bolded, and the next best model is underlined.

| | | FI-2010 | | | | |
|---|---|---|---|---|---|---|
| Model | Metric | K=1 | K=2 | K=3 | K=5 | K=10 |
| MLP | MSE | 0.659 (0.005) | 1.096 (0.021) | 1.431 (0.020) | 1.972 (0.008) | 2.963 (0.162) |
| | Corr | 0.084 (0.002) | 0.101 (0.001) | 0.102 (0.011) | 0.106 (0.006) | 0.125 (0.019) |
| | $R^2$ | -0.004 (0.007) | -0.016 (0.020) | -0.016 (0.014) | -0.008 (0.004) | -0.060 (0.058) |
| LSTM | MSE | **0.638 (0.005)** | **1.035 (0.009)** | **1.328 (0.007)** | **1.824 (0.013)** | **2.571 (0.024)** |
| | Corr | **0.173 (0.020)** | **0.211 (0.024)** | **0.244 (0.007)** | **0.274 (0.014)** | 0.298 (0.009) |
| | $R^2$ | **0.027 (0.007)** | **0.041 (0.008)** | **0.057 (0.005)** | **0.067 (0.006)** | **0.081 (0.009)** |
| CNN1 | MSE | 0.665 (0.008) | 1.058 (0.007) | 1.379 (0.015) | 1.879 (0.023) | 2.681 (0.017) |
| | Corr | 0.129 (0.006) | 0.185 (0.013) | 0.210 (0.004) | 0.248 (0.016) | **0.298 (0.007)** |
| | $R^2$ | -0.013 (0.012) | 0.020 (0.007) | 0.021 (0.011) | 0.039 (0.012) | 0.041 (0.006) |
| BINCTABL | MSE | 0.650 (0.000) | 1.047 (0.001) | 1.347 (0.008) | 1.838 (0.016) | 2.612 (0.012) |
| | Corr | 0.106 (0.004) | 0.176 (0.003) | 0.215 (0.015) | 0.249 (0.016) | 0.278 (0.003) |
| | $R^2$ | 0.010 (0.000) | 0.029 (0.001) | 0.044 (0.006) | 0.061 (0.008) | 0.069 (0.004) |
| DAIN | MSE | 0.693 (0.011) | 1.114 (0.014) | 1.436 (0.004) | 1.977 (0.004) | 2.824 (0.014) |
| | Corr | 0.038 (0.004) | 0.068 (0.007) | 0.085 (0.002) | 0.107 (0.003) | 0.127 (0.001) |
| | $R^2$ | -0.057 (0.016) | -0.032 (0.013) | -0.019 (0.003) | -0.011 (0.002) | -0.007 (0.005) |
| TRANSLOB | MSE | 0.659 (0.005) | 1.088 (0.002) | 1.395 (0.003) | 1.904 (0.015) | 2.704 (0.029) |
| | Corr | 0.079 (0.010) | 0.090 (0.019) | 0.150 (0.018) | 0.210 (0.004) | 0.267 (0.009) |
| | $R^2$ | -0.005 (0.008) | -0.008 (0.002) | 0.009 (0.002) | 0.027 (0.008) | 0.033 (0.010) |
| PatchTST | MSE | 0.654 (0.000) | 1.077 (0.000) | 1.406 (0.001) | 1.949 (0.001) | 2.795 (0.002) |
| | Corr | 0.081 (0.002) | 0.082 (0.001) | 0.079 (0.004) | 0.092 (0.003) | 0.085 (0.003) |
| | $R^2$ | 0.003 (0.001) | 0.002 (0.000) | 0.002 (0.001) | 0.004 (0.000) | 0.001 (0.001) |
| DLinear | MSE | 0.652 (0.000) | 1.073 (0.000) | 1.402 (0.000) | 1.945 (0.001) | 2.782 (0.000) |
| | Corr | 0.080 (0.002) | 0.081 (0.001) | 0.074 (0.001) | 0.083 (0.002) | 0.084 (0.002) |
| | $R^2$ | 0.006 (0.000) | 0.006 (0.000) | 0.005 (0.000) | 0.006 (0.001) | 0.005 (0.000) |
| iTransformer | MSE | 0.683 (0.008) | 1.183 (0.031) | 1.582 (0.016) | 2.279 (0.095) | 3.401 (0.076) |
| | Corr | 0.045 (0.004) | 0.045 (0.007) | 0.033 (0.005) | 0.063 (0.004) | 0.056 (0.004) |
| | $R^2$ | -0.041 (0.012) | -0.096 (0.028) | -0.123 (0.012) | -0.165 (0.048) | -0.216 (0.027) |
| TimeMixer | MSE | 0.657 (0.001) | 1.075 (0.002) | 1.394 (0.002) | 1.888 (0.018) | 2.643 (0.017) |
| | Corr | 0.083 (0.005) | 0.110 (0.001) | 0.135 (0.005) | 0.201 (0.025) | 0.271 (0.014) |
| | $R^2$ | -0.001 (0.002) | 0.004 (0.001) | 0.011 (0.001) | 0.035 (0.009) | 0.055 (0.006) |

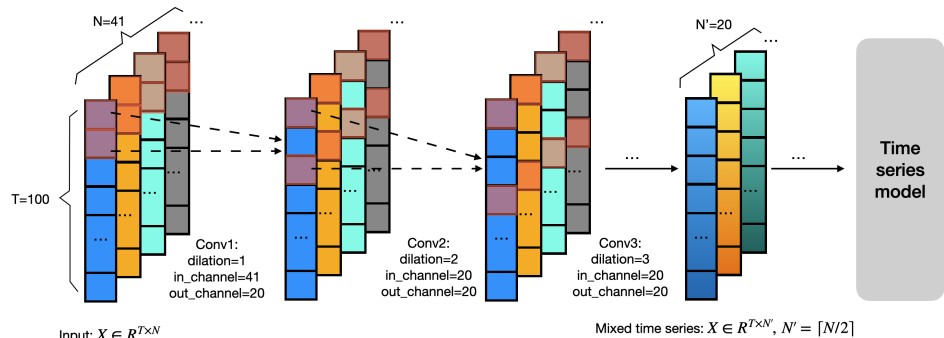

Figure 2: **Cross-Variate Mixing Layers (CVML) Architecture.** CVML consists of 5 Conv1D layers, with the first 3 illustrated here. Each Conv1D layer employs a kernel size of 2 and matches its input channels to the number of variates in the input time series. The architecture features increasing dilation across successive layers to expand the receptive field. Input $X \in \mathbb{R}^{T \times N}$ is transformed into a mixed time series $X' \in \mathbb{R}^{T \times N'}$, $N' = \lceil N/2 \rceil$, before feeding into the subsequent time series model.

Table 5: **Time Series Model Performance with and without CVML** on the FI-2010 Dataset using basic LOB features. The % column indicates the percentage improvement from adding CVML.

| Model | MSE (↓) | | | | | % | Corr (↑) | | | | | % | $R^2$ (↑) | | | | | % |
|---|---|---|---|---|---|---|---|---|---|---|---|---|---|---|---|---|---|---|
| | K=1 | K=2 | K=3 | K=5 | K=10 | | K=1 | K=2 | K=3 | K=5 | K=10 | | K=1 | K=2 | K=3 | K=5 | K=10 | |
| PatchTST-CVML | **0.653** | **1.071** | **1.370** | **1.893** | **2.646** | 3.1 | **0.070** | **0.113** | **0.165** | **0.191** | **0.241** | 86.2 | **0.005** | **0.007** | **0.028** | **0.033** | **0.054** | 958.3 |
| PatchTST | 0.654 | 1.077 | 1.406 | 1.949 | 2.795 | | 0.081 | 0.082 | 0.079 | 0.092 | 0.085 | | 0.003 | 0.002 | 0.002 | 0.004 | 0.001 | |
| DLinear-CVML | **0.650** | **1.042** | **1.352** | **1.796** | **2.548** | 5.9 | **0.104** | **0.192** | **0.205** | **0.291** | **0.313** | 174.9 | **0.010** | **0.035** | **0.040** | **0.082** | **0.089** | 814.3 |
| DLinear | 0.652 | 1.073 | 1.402 | 1.945 | 2.782 | | 0.080 | 0.081 | 0.074 | 0.083 | 0.083 | | 0.006 | 0.006 | 0.005 | 0.006 | 0.005 | |
| iTransformer-CVML | **0.654** | 1.084 | **1.402** | **2.002** | **2.649** | 14.6 | **0.054** | **0.070** | **0.088** | **0.121** | **0.249** | 140.5 | **0.002** | **-0.005** | **0.005** | **-0.024** | **0.053** | 104.8 |
| iTransformer | 0.683 | **1.183** | 1.582 | 2.279 | 3.401 | | 0.045 | 0.045 | 0.033 | 0.063 | 0.056 | | -0.041 | -0.096 | -0.123 | -0.165 | -0.216 | |
| TimeMixer-CVML | **0.642** | **1.033** | **1.329** | **1.807** | **2.494** | 4.6 | **0.160** | **0.221** | **0.257** | **0.298** | **0.353** | 61.1 | **0.022** | **0.043** | **0.056** | **0.076** | **0.109** | 194.2 |
| TimeMixer | 0.657 | 1.075 | 1.394 | 1.888 | 2.643 | | 0.083 | 0.110 | 0.135 | 0.201 | 0.271 | | -0.001 | 0.004 | 0.011 | 0.035 | 0.055 | |

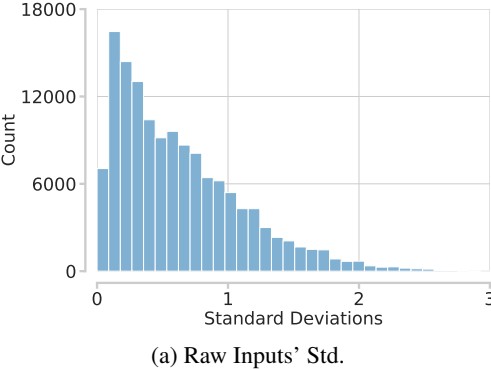
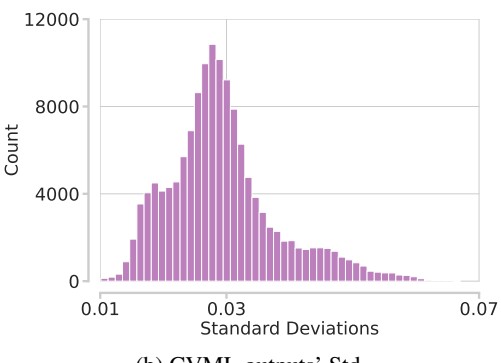

(a) Raw Inputs' Std.    (b) CVML outputs' Std.

Figure 3: **Standard Deviation Distribution (Std.) of Inputs Before and After CVML Processing on the FI-2010 test set.** This histogram is from the TimeMixer with CVML add-on. The CVML outputs show lower std values and a distribution closer to normal, suggesting noise reduction.

CVML's role in improving the signal-to-noise ratio of LOB data, thereby facilitating more accurate predictions by subsequent time series models.

We demonstrate that CVML enhances the ability of time series models to capture cross-variate and temporal correlations, using iTransformer and PatchTST as examples. We chose them for their interpretable attention mechanisms and their representative modeling on different correlation types: iTransformer focuses on cross-variate correlations through self-attention on the variate dimension, while PatchTST emphasizes temporal correlations via self-attention on time dimension patches. We train both models on FI-2010 and analyze the average attention scores from their final attention layers across the test set. For iTransformer, we visualize attention scores from the mid-price return (target variate, id: 40) to all other variates and itself. For PatchTST, we examine average attention scores from the last time step patch (id: 10) to all other patches and itself.

**Cross-Variate Correlations:** Figure 4a reveals that without CVML, iTransformer's mid-price return attention is predominantly self-focused, with a uniform pattern corresponding to the LOB input feature layout ($\{V_i^{\text{bid}}, P_i^{\text{bid}}, V_i^{\text{ask}}, P_i^{\text{ask}}\}_{i=1}^{10}$). This uniform attention suggests only surface-level capture of cross-variate correlations, failing to differentiate between LOB levels. Notably, it doesn't reflect the expected stronger correlation of mid-price to the best bid and ask prices. In contrast, Figure 4b shows that with CVML, iTransformer exhibits varied attention across variates, indicating a more nuanced modeling of LOB-level correlations. **Temporal Correlations:** Figure 4c demonstrates PatchTST's attention w/o CVML, showing strong self-attention for the latest time patch but no discernible temporal pattern with other patches. Conversely, Figure 4d demonstrates that with CVML, PatchTST captures a clear decaying temporal pattern and stronger attention to immediate past neighbors, indicating improved modeling of temporal dependencies.

### 3.4 ABLATION STUDY

CVML is designed to capture two types of correlations: cross-variate and temporal. To demonstrate the efficacy of this design, we conduct an ablation study using two modified versions of CVML. The first variant (CVML-abla1) reduces the kernel size to 1, focusing solely on cross-variate correlations

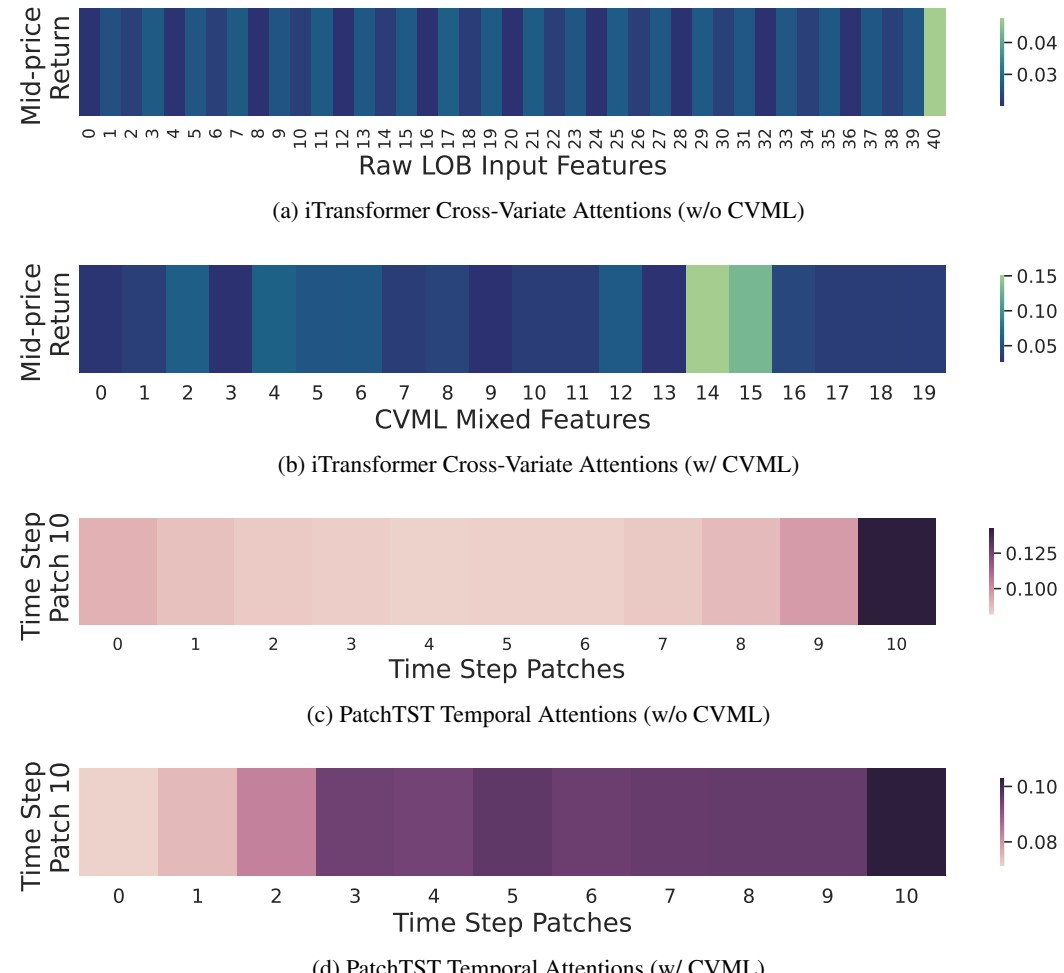

Figure 4: **Impact of CVML on Cross-Variate and Temporal Attention Scores.** (a, b) Cross-variate attention from mid-price return (id: 40) to all variates. (c, d) Temporal attention from the latest time step patch (id: 10) to all historical patches. iTransformer w/ CVML (b) captures more nuanced cross-variate correlations compared to (a). PatchTST w/ CVML (d) reveals clearer temporal dependencies than (c).

while eliminating temporal correlations. The second variant (CVML-abla2) maintains the original kernel size of 2 but employs depthwise convolution (Chollet, 2017; Pandey, 2024), which processes each variate independently without cross-variate information aggregation. We replicate the experiments from Table 5 using these ablated versions and compare their average forecast performance across the five prediction horizons. Figure 5 shows that both ablated versions underperform the original CVML. These results underscore the importance of CVML's dual-correlation design, highlighting its ability to effectively capture both cross-variate and temporal dependencies in LOB data. The full results of the two ablated CVMLs including MSE, Corr, and $R^2$ are in Table 14.

To verify that CVML's performance gain does not come model size increase, we examine the model size of each model before and after adding CVML. Table 6 shows the number of learnable parameters. The percentage indicates the size of the vanilla model compared to the counterpart with CVML. Except TimeMixer, all other models are of more than 90% size of the counterpart with CVML. Thus, we increase TimeMixer's number of layers to increase its learnable parameters to 14156, about 109% of the CVML version and test its performance.

To ensure that CVML's performance gains are not solely attributable to increased model complexity, we compare the model sizes before and after incorporating CVML. Table 6 presents the number of learnable parameters for each model, with percentages indicating the size of the vanilla model relative to its CVML-enhanced counterpart. All models except TimeMixer is over 90% of the size of the version with CVML integration. We augment TimeMixer's architecture by increasing its number of layers, resulting in 14,156 learnable parameters, approximately 109% of its CVML version's size.

We then evaluate this enlarged TimeMixer and it still significantly underperforms TimeMixer-CVML, proving that CVML's gains are not solely attributable to increased model complexity.

Table 6: Model Size Comparison

|  | PatchTST | DLinear | iTransformer | TimeMixer |
|---|---|---|---|---|
| Vanilla | 33766 (92.65%) | 8282 (148%) | 6358017 (99.96%) | 9471 (72.71%) |
| w/ CVML | 36444 | 5614 | 6360803 | 13025 |

Table 7: Enlarged TimeMixer (109% of TimeMixer+CVML size)

|  | K=1 | K=2 | K=3 | K=5 | K=10 |
|---|---|---|---|---|---|
| Corr | 0.072 | 0.110 | 0.146 | 0.206 | 0.268 |
| $R^2$ | -0.010 | -0.001 | 0.011 | 0.030 | 0.053 |

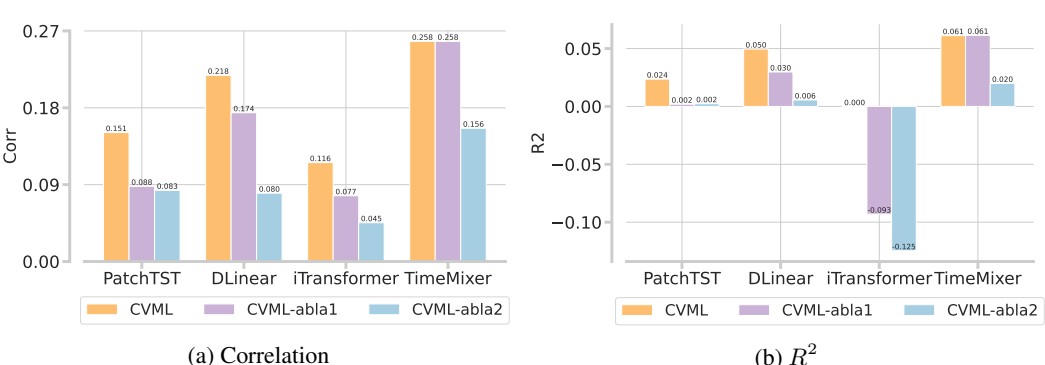

(a) Correlation

(b) $R^2$

Figure 5: **Average Correlation and $R^2$ Scores Across Five Prediction Horizons.** This figure compares the performance of CVML and its two ablated versions (CVML-abla1 and CVML-abla2) across four time series models. The CVML consistently outperforms its ablated counterparts. Notably, CVML-abla2, which lacks cross-variate information aggregation, performs the worst, highlighting the critical importance of cross-variate mixing in CVML's effectiveness.

## 4 CONCLUSION

In LOBBen-TM, we build a unified evaluation pipeline that progresses from trends to returns and from features to models. We first standardize a full LOB feature set and run a cross-asset MPTP benchmark on FI-2010 (equities) and Bitcoin (crypto) to test generalization under a common protocol. We then open the black box of the feature pipeline by benchmarking a feature taxonomy (basic, time-insensitive, time-sensitive) and conducting a targeted FI-2010 ablation, which shows that time-sensitive features yield consistent, sizable gains—pointing to the centrality of temporal signal modeling in LOB prediction. Motivated by this result, we extend the scope from trend classification to MPRF and jointly evaluate LOB-specific architectures and top-tier general time-series (TS) backbones on FI-2010. Finally, to strengthen multivariate interactions in TS backbones on LOB returns, we introduce a lightweight Cross-Variate Mixing Layer (CVML) that plugs into existing models. Our research yields **four** conclusions:

- **Feature Importance.** Across MPTP, the common LOB feature taxonomy (basic, time-insensitive, time-sensitive) is predictive; on FI-2010, *time-sensitive* features deliver consistent, sizable gains, underscoring the value of explicit temporal modeling.
- **Cross-Asset Generalization.** Using the standardized *full* feature set, MPTP performance is robust on both FI-2010 and Bitcoin; however, model rankings shift across assets, highlighting heterogeneous LOB dynamics and the limits of architecture-only transfer.
- **Model Specialization for MPRF.** On FI-2010 MPRF, LOB-specific inductive biases remain advantageous over general TS backbones under standard metrics (MSE, $R^2$, Corr), indicating that returns forecasting benefits from LOB-aware design.
- **CVML Plug-in Benefit.** The proposed CVML substantially improves general TS models on MPRF, narrowing the gap to LOB-specific designs and offering a simple, effective path to stronger multivariate mixing on LOB data.

ETHICS STATEMENT

We have reviewed the Code of Ethics and affirm that all authors have read it, adhere to it, and that this submission complies with its requirements.

REPRODUCIBILITY STATEMENT

We have provided all hyperparameters to reproduce the results. The code to reproduce the results of this paper is accessible at this link: https://tinyurl.com/49s8bb6n

The data we use can be downloaded from these links: FI-2010: https://tinyurl.com/5cbzxnnc, Bitcoin: https://tinyurl.com/mv68vheb

LLM USAGE DISCLOSURE

We used large language models (LLMs) to aid and polish writing, such as improving clarity, grammar, and conciseness. We also used LLMs for retrieval and discovery, for example exhausting literature to identify potential missing related work. All technical content, proofs, experiments, and results are original contributions by the authors.

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

# Appendix

## A   RELATED WORKS

**Price Trend Prediction Surveys.** Several comprehensive benchmark surveys have examined deep learning applications in price trend prediction (Ozbayoglu et al., 2020; Sezer et al., 2020; Jiang, 2021), each with a distinct focus. Jiang (2021) emphasize reproducibility, analyzing model architectures, evaluation metrics, and implementations in stock price and market index prediction studies from 2017 to 2019. A follow-up study by Kumbure et al. (2022) extend this analysis to datasets and input variables commonly used in stock market predictions. Hu et al. (2021) review 86 papers on stock and foreign exchange price prediction, while other surveys (Rundo et al., 2019; Mintarya et al., 2023) compare machine learning and deep learning methods in stock market prediction, concluding that deep learning approaches generally offer superior accuracy. Nti et al. (2020) broaden the scope beyond technical analysis, reviewing 122 papers from 2007 to 2018 covering technical, fundamental, and combined analyses. Additionally, Shah et al. (2019) evaluate the real-world applicability of models through backtesting performance. Notably, these surveys do not include benchmarks for prediction models on Limit Order Book (LOB) data. The most relevant work is a benchmark study by Prata et al. (2024) on mid-price trend prediction models using LOB data. However, our work addresses three key limitations of their study: We use a proprietary dataset with a time range 200 times larger than the open-source dataset they employed. We benchmark models on both stock and Bitcoin datasets, whereas they focused solely on stocks. We extend our analysis to include the mid-price forecasting problem (a regression task), benchmarking both LOB models and state-of-the-art time series forecasting models, in addition to the mid-price trend prediction task (a classification problem) they addressed. These enhancements allow our study to provide a more comprehensive and diverse evaluation of LOB-based prediction models across different assets and problem types.

**Time Series Forecasting Models.** The success of deep learning in natural language processing and computer vision has significantly influenced time series forecasting, with deep learning models becoming predominant in this field. Transformer-based architectures, in particular, have emerged as the leading approach for multivariate time series forecasting (Nie et al., 2022; Liu et al., 2023). However, recent developments have shown that models based on linear layers (Zeng et al., 2023; Wang et al., 2024; Chen et al., 2023; Oreshkin et al., 2020; Challu et al., 2023; Zhang et al., 2022) can achieve comparable performance to transformer-based models. While convolutional neural networks (O'shea & Nash, 2015; Wu et al., 2023; Franceschi et al., 2019) and recurrent networks (Hochreiter & Schmidhuber, 1997; Lai et al., 2018; Franceschi et al., 2019) have also been applied to time series forecasting, their performance generally lags behind that of transformers and linear-based architectures. Notably, there has been limited overlap between general time series forecasting and Limit Order Book (LOB) time series analysis. To date, no comprehensive benchmarking of state-of-

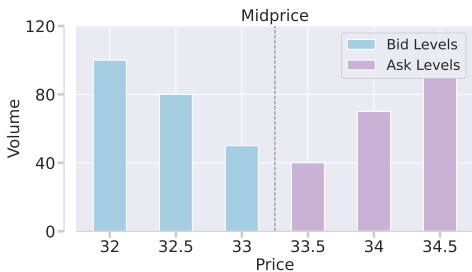

Figure 6: **Visualizing Limit Order Book Data:** three *bid* and three *ask* levels of varying price and volume are shown, as well as the *mid-price*.

the-art time series forecasting models on the complex LOB time series data has been conducted. To address this gap, our paper selects four state-of-the-art time series forecasting models, encompassing both transformer-based and linear architectures. Our aim is to bridge the divide between general-purpose time series forecasting and the more specialized field of LOB time series forecasting, providing insights into the applicability and performance of these models on LOB data.

## B  BACKGROUND ON LIMIT ORDER BOOK (LOB)

Global exchanges use matching engines to pair orders from bid and ask sides of market participants. The Limit Order Book is the essential data structure organizing these orders, reflecting market supply and demand. Three common order types exist: **1)** *Market orders* are requests to buy or sell a specified number of shares at the best available price, usually executed immediately. **2)** *Limit orders* are requests to buy or sell a specified number of shares at a specified price, often queued for matching due to price constraints. **3)** *Cancel orders* are requests to withdraw previously submitted limit orders.

## C  DETAILED INFORMATION OF DATASETS AND MODELS IN BENCHMARK

### C.1  MODELS FOR MID-PRICE TREND PREDICTION

- **MLP, LSTM, CNN1, CNN2, CNNLSTM.** Tsantekidis et al. (2017b) used Multilayer Perceptron (MLP) and Long Short-Term Memory (LSTM) to predict future mid-price movements. The same authors (Tsantekidis et al., 2017a) also proposed a Convolutional Neural Network (CNN) model (CNN1), which employs a standard CNN architecture with convolutional layers followed by fully connected layers. In 2020, the same research team (Tsantekidis et al., 2020) proposed another CNN model (CNN2) and a CNNLSTM model based on the described LSTM and CNN2. The key improvement of CNN2 is the use of causal convolutions with "full" padding, ensuring all convolutional layers produce outputs with the same number of time steps. This matches labels to their correct time steps and prevents future data from influencing past predictions. The CNNLSTM model merges CNN2 and LSTM, using CNN2 for feature extraction and passing the features to the LSTM module for classification. In our experiments, we adjusted the CNN kernel size to match the different number of features in each dataset.
- **DAIN.** Passalis et al. (2019) introduce Deep Adaptive Input Normalization (DAIN) for time series forecasting. The main innovation of the method is learning to adaptively normalize input data during model training via two feed-forward layers, an adaptive shifting layer and scaling layer, instead of using fixed schemes like z-score normalization. Besides, they use a gating layer to suppress irrelevant features. DAIN explores three possible neural network architectures: MLP, CNN, and RNN. We choose the MLP architecture as it demonstrates the highest empirical performance.
- **CTABL, BINCTABL.** Tran et al. (2018) propose the Temporal-Attention-Augmented Bilinear Layer (TABL) model. The Bilinear Layer (BL) performs two linear transformations on the input data along the feature and temporal dimensions to capture how stock prices interact at a given point in time and how prices at an index progress over time. However, within a BL, it is unclear how different time instance representations interact. TABL addresses this by incorporating a temporal attention mechanism into the BL model to learn the importance of each time instance relative to others. The input time series is first passed through the feature dimension transformation,

then through the temporal attention mechanism, and finally through the temporal dimension transformation. In our experiments, we use the C(TABL) variant of TABL for its superior empirical performance over A(TABL) and B(TABL). Tran et al. (2021) improve upon TABL with BINCTABL, which incorporates a Bilinear Normalization (BiN) strategy that normalizes the data along the temporal and feature dimensions with learnable parameters to scale the normalization. The authors compare BiN as a simpler and more intuitive approach compared to DAIN when using TABL networks.

- **DEEPLOB.** Zhang et al. (2019) propose Deep Convolutional Neural Networks for Limit Order Books (DeepLOB). The authors employ a mid-price-based smooth data labeling method to reduce noise and eliminate minor oscillations. DEEPLOB combines convolutional layers and an Inception Module to extract features from the noisy financial data, then uses a Long Short-Term Memory (LSTM) layer to capture longer time dependencies among the extracted features. In our experiments, we adjust the kernel size of the last convolution layer for different datasets.

- **DEEPLOBATT.** Zhang & Zohren (2021) introduce DeepLOB-Attention, an encoder-decoder model built upon their previous work, DEEPLOB. DEEPLOBATT utilizes DEEPLOB as the encoder and an Attention model (Luong et al., 2015) as the decoder. The authors investigated both Attention and Seq2Seq as the decoder in their paper. Since DEEPLOBATT outperforms DEEPLOB-Seq2Seq in nearly all experiments, we choose the DEEPLOBATT model in our experiments and adjust the kernel size of the convolutional block for different datasets according to the feature dimensions.

- **TLONBoF.** Passalis et al. (2020) propose Temporal Logistic Neural Bag-of-Features (TLo-NBoF) (2020), a refined Bag-of-Features (BoF) formulation to better capture the dynamics of time series data compared to existing BoF methods. They achieved this by introducing a novel adaptive scaling mechanism and replacing the Gaussian density estimation of regular BoF with a logistic kernel. The input time series is first passed through a 1-D convolution layer to extract relationships between time instances. Then, the TLONBoF formulation uses a learnable logistic kernel and codebook to aggregate the feature vectors into short-term, mid-term, and long-term histograms that capture the overall behaviors of the input time series. This adaptive method removes the need for sophisticated initialization methods and facilitates smoother hyperparameter tuning.

- **DLA.** Guo & Chen (2022) propose a Dual-Stage Temporal Attention-Based Deep Learning Architecture (DLA). It uses a dual-stage temporal attention mechanism to repeatedly highlight the most important time instances in input time series data. In the first stage, temporal attention is performed on the input data to learn the importance of each time instance relative to the others. The output of the attention is passed through a stacked Gated Recurrent Unit (GRU) network to further enhance the representational state of the input. In the second stage, temporal attention weights are adaptively assigned to the hidden states of the GRU network. The model's performance is benchmarked against other models in the literature, such as CTABL (Tran et al., 2018), DEEPLOB (Zhang et al., 2019), and TLONBoF (Passalis et al., 2020).

- **TRANSLOB.** Wallbridge (2020) proposes TransLOB, a new model architecture for LOB data that uses Transformer blocks (Vaswani et al., 2023). The TRANSLOB architecture consists of a causal convolution module and a causal transformer block to stay consistent with the nature of time-series data. The convolutional module comprises 5 1-D convolution layers with increasing dilation to capture relationships between short-term and long-term time instances. The transformer block includes 2 masked self-attention encoders to determine important time instance representations. The performance of TRANSLOB is benchmarked against other state-of-the-art models such as CTABL, DEEPLOB, and CNN-LSTM.

## C.2 MODELS FOR MID-PRICE FORECASTING

Besides the LOB Models including MLP, CNN1, LSTM, BINCTABL, DAIN, and TRANSLOB mentioned above, we also include 4 state-of-the-art time series forecasting models.

- **PatchTST.** Nie et al. proposes PatchTST (Nie et al., 2022), a multivariate time series forecasting model based on vanilla Transformer architecture. It has two major novelties. First, it uses the same model to predict each variate of a multivariate time series model independently, making it technically a univariate time series model. Second, to better capture time series' local pattern, it models the time series in patch-wise (each patch includes multiple consecutive time steps) rather than timestep-wise.

- **iTransformer.** To better capture variate-centric information rather than only temporal dimension, Liu et al. (2023) proposes iTransformer. It uses the vanilla transformer components in a novel way where it first embeds each series of a multivariate time series into a token and then applies attention across all the tokens representing different series.
- **DLinear.** Besides transformer-based models, linear models are also showing top performance in time series forecasting. Zeng et al. (2023) questions Transformer architecture's capability to properly capture temporal correlations in time series data. They propose a simple baseline architecture, DLinear. It is a one-layer linear model with a decomposition component. It first decomposes the raw time series input into a trend component and a seasonal component. Then, two separate one-layer linear layers are applied on these two components. Lastly, two components sum up in to the final prediction. DLinear outperformans most of the Transformer-based time series prediction model and achieve top-tier time series forecasting performance.
- **TimeMixer.** Along the line of better modeling temporal correlations from the time series data, Wang et al. (2024) proposes TimeMixer. It is a linear-layer-based model. Its major novelty is to first preprocess the raw input time series into multiple time series of different resolutions by downsampling. Then, they mix pairs of time series of different resolutions by adding one of the time series to a transformation of the other. The transformation consists of two linear layers and an intermediate GELU activation function. Lastly, they use a linear layer to regress from the mixed time series to produce the final prediction.

## D HYPERPARAMETERS

Table 8, and Table 9 include the hyperparameters we use in MPTP and MPRF. For Mid-Price Trend Prediction (MPTP), we adhere to the hyperparameters specified in the original papers for the models. We apply a patience of 3 for MPTP on both datasets. For Mid-Price Return Forecasting (MPRF), we perform a grid search for the batch size, including {32, 64, 128} and the learning rate, including {0.01, 0.001, 0.0001, 0.00001} on horizon 5. We employ a patience of 15.

Table 8: Hyperparameters for Mid-Price Trend Prediction. Dashes mean the corresponding module does not exist in the model architecture.

| Model | Learning Rate | Optimizer | Batch Size | Epochs | Dropout | MLP Hidden | RNN Hidden |
|---|---|---|---|---|---|---|---|
| LSTM | 0.001 | Adam | 32 | 100 | 0 | 64 | 40 |
| MLP | 0.001 | Adam | 64 | 100 | 0.1 | 128 | - |
| CNN1 | 0.0001 | Adam | 64 | 100 | - | 32 | - |
| CTABL | 0.01 | Adam | 256 | 200 | 0.1 | - | - |
| DAIN | 0.0001 | RMSprop | 32 | 100 | 0.5 | 512 | - |
| DEEPLOB | 0.01 | Adam | 32 | 100 | - | - | 64 |
| CNNLSTM | 0.001 | RMSprop | 32 | 100 | 0.1 | 32 | 32 |
| CNN2 | 0.001 | RMSprop | 32 | 100 | - | 32 | - |
| TRANSLOB | 0.0001 | Adam | 32 | 150 | 0.1 | 64 | - |
| TLONBoF | 0.0001 | Adam | 128 | 100 | 0.5 | 512 | - |
| BINCTABL | 0.001 | Adam | 128 | 200 | 0.1 | - | - |
| DEEPLOBATT | 0.001 | Adam | 32 | 100 | - | - | 64 |
| DLA | 0.01 | Adam | 256 | 100 | 0.5 | - | 100 |

Table 9: Hyperparameters for Mid-Price Return Forecasting on the FI-2010 Dataset

| Model | Learning Rate | Optimizer | Batch Size | Epochs | Dropout | MLP Hidden | RNN Hidden | Transformer Hidden |
|---|---|---|---|---|---|---|---|---|
| MLP | 0.0001 | Adam | 32 | 50 | 0.1 | 128 | - | - |
| LSTM | 0.01 | Adam | 64 | 50 | 0.2 | 64 | 32 | - |
| CNN1 | 0.0001 | Adam | 32 | 50 | - | - | 32 | - |
| BINCTABL | 0.0001 | Adam | 64 | 50 | 0.1 | 128 | 32 | - |
| DAIN | 0.0001 | Adam | 128 | 50 | 0.5 | - | - | - |
| TRANSLOB | 0.0001 | Adam | 128 | 50 | 0.1 | 64 | - | 60 |
| PATCHTST | 0.0001 | Adam | 128 | 50 | 0.3 | - | - | 128 |
| DLINEAR | 0.0001 | Adam | 128 | 50 | - | - | - | - |
| ITRANSFORMER | 0.0001 | Adam | 128 | 50 | 0.1 | - | - | 512 |
| TIMEMIXER | 0.0001 | Adam | 128 | 50 | 0.1 | 16 | - | - |

# E    ADDITIONAL MPRF RESULTS ON MULTIVARIATE SYNTHETIC DATASETS

To further demonstrate CVML's ability to capture cross-variate correlation, we generate a synthetic dataset and test CVML's performance on it. Compared to real datasets with latent cross-variate correlations such as FI-2010 and the bitcoin dataset, the synthetic dataset has well-defined cross-variate correlations specified in the data generation code. The synthetic dataset's target is a synthetic electricity price, the other variates are electricity load, electricity production and temperature. We generate the data in a way that the temperature affects the electricity load (e.g. cold and hot temperature increase the electricity load), the electricity load and the production affect the electricity price (e.g. higher load increases the price and higher production lower the price).

## E.1    SYNTHETIC TIME SERIES DATA GENERATION

The synthetic dataset consists of multiple time series components with complex relationships and non-linear interactions. The data generation process follows a hierarchical structure where intermediate variables influence the final target variable (electricity price). There are totally six varieties: electricity price, load, production, temperature, supply_margin, price_volatility. These varieties have cross-variate correlations. For example, an increasing load of electricity leads to an increasing electricity price. An increasing production of electricity leads to a decreasing electricity price. When the temperature is too high or too cold, the load will increase. We introduce the detailed generation process of each variate as follows.

**Temperature Generation**: Let $t \in \{0, 1, ..., n-1\}$ represent the time index for $n$ hourly observations. The temperature time series $T_t$ is generated as:

$$T_t = 20 + 10\sin(()\text{seasonal\_cycle}) + 5\sin(()\text{daily\_cycle}) + \epsilon_T + \delta_H - \delta_C \tag{E.1}$$

where:

- seasonal_cycle $= \frac{2\pi t}{24 \times 365.25}$
- daily_cycle $= \frac{2\pi t}{24}$
- $\epsilon_T \sim \mathcal{N}(0, 2^2)$ represents random fluctuations
- $\delta_H \sim \text{Bernoulli}(0.05) \times 8$ represents heat waves
- $\delta_C \sim \text{Bernoulli}(0.05) \times 8$ represents cold snaps

**Load Generation**: The electricity load $L_t$ is modeled as (including base load, daily pattern, AC usage, heating, seasonal pattern, weekday effect, and a random noise):

$$L_t = 1000 + 200\sin(()\text{daily\_cycle}) + 150\mathbb{1}_{T_t > 25} + 100\mathbb{1}_{T_t < 10} + 100\sin(()\text{seasonal\_cycle}) + 50\mathbb{1}_{\text{weekday}} + \epsilon_L \tag{E.2}$$

where $\epsilon_L \sim \mathcal{N}(0, 30^2)$ and $\mathbb{1}$ represents the indicator function.

**Production Generation**: The production capacity $P_t$ is generated through multiple components:

$$P_t = (1.1L_t + 100\sin(()\text{daily\_cycle}) + \epsilon_P) \times M_t \times O_t \tag{E.3}$$

where:

- $\epsilon_P \sim \mathcal{N}(0, 30^2)$
- $M_t \sim \text{Categorical}([1.0, 0.7], [0.9, 0.1])$ represents maintenance periods
- $O_t \sim \text{Categorical}([1.0, 0.3], [0.98, 0.02])$ represents outages

**Electricity Price Generation (Target Variable)**: The final price $Y_t$ is generated through a complex interaction of components:

$$Y_t = ((50 + 0.08L_t - 0.04P_t + 0.7(T_t - 20)^2 + 15\sin(()\text{daily\_cycle}) + 10\sin(()\text{seasonal\_cycle})) \times R_t \times S_t \times D_t + \epsilon_Y) \tag{E.4}$$

where:

- $R_t \sim \text{Categorical}([1.0, 1.5, 0.7], [0.7, 0.15, 0.15])$ represents regime changes

Table 10: **Time Series Model Performance with and without CVML** on the Synthetic Dataset using basic LOB features. The % column indicates the percentage improvement from adding CVML.

| Model | MSE ($\downarrow$) | | | | Corr ($\uparrow$) | | | | $R^2$ ($\uparrow$) | | | |
|---|---|---|---|---|---|---|---|---|---|---|---|---|
| | K=1 | K=5 | K=10 | % | K=1 | K=5 | K=10 | % | K=1 | K=5 | K=10 | % |
| PatchTST-CVML | **0.807** | **0.7538** | **0.7589** | 26 | **0.4298** | **0.485** | **0.48** | 18.1 | **0.1818** | **0.2346** | **0.2296** | 359.7 |
| PatchTST | 1.0508 | 1.0441 | 1.0382 | | 0.3907 | 0.3915 | 0.3991 | | -0.0654 | -0.0602 | -0.054 | |
| DLinear-CVML | **0.7825** | **0.7733** | **0.8743** | 8.6 | **0.4585** | **0.4639** | **0.3364** | 31.7 | **0.2067** | **0.2147** | **0.1124** | 76.2 |
| DLinear | 0.8861 | 0.8847 | 0.8868 | | 0.3188 | 0.3209 | 0.3159 | | 0.1016 | 0.1017 | 0.0997 | |
| iTransformer-CVML | **0.8544** | **0.8183** | **0.8179** | 28.8 | **0.4011** | **0.4235** | **0.4317** | 18 | **0.1337** | **0.1691** | **0.1697** | 85.9 |
| iTransformer | 1.1667 | 1.161 | 1.1706 | | 0.339 | 0.3627 | 0.3628 | | -0.1829 | -0.1789 | -0.1884 | |
| TimeMixer-CVML | **0.7469** | 0.7704 | **0.7539** | 1 | **0.493** | 0.4693 | **0.4865** | 2.4 | **0.2427** | 0.2177 | **0.2347** | 4.4 |
| TimeMixer | 0.7622 | **0.7596** | 0.7781 | | 0.4774 | **0.4782** | 0.4586 | | 0.2272 | **0.2287** | 0.2101 | |

- $S_t$ represents price spikes triggered by conditions:

$$S_t = \begin{cases} \sim \text{Categorical}([1.0, 2.5], [0.7, 0.3]) & \text{if } C_t = 1 \\ 1.0 & \text{otherwise} \end{cases}$$

where $C_t = 1$ if any of the following conditions are met:

  - $L_t/P_t > 0.9$ (high demand relative to production)
  - $T_t > 30$ (very hot weather)
  - $T_t < 0$ (very cold weather)
  - $P_t/P_{\text{base},t} < 0.5$ (significant production issues)
- $D_t \sim \text{Categorical}([1.0, 0.4], [0.97, 0.03])$ represents sudden price drops
- $\epsilon_Y \sim \mathcal{N}(0, (0.3 \times 100)^2)$ represents price noise

### E.1.1 DERIVED FEATURES

Additional features are computed from the primary variables:

$$\text{supply\_margin}_t = \frac{P_t - L_t}{L_t}$$

$$\text{price\_volatility}_t = \sigma(\{Y_{t-23}, ..., Y_t\})$$

where $\sigma$ represents the rolling standard deviation over a 24-hour window.

As shown in Table 10, this synthetic time series dataset, although it was defined in a specific domain (electricity) for better interpretability, includes generic patterns that could be found across different domains of multivariate time series. Specifically, it includes the following patterns: multiple seasonality (e.g. weekly/monthly sales cycles, weekday/weekend differences in web traffic), non-linear relationships, temporary anomalies and recoveries (e.g. electricity outage), periods of high volatility followed by calmer periods (e.g. viral content spread on social media), multi-factor interactions (e.g. inventory-price-demand relationships in supply chain) and stochastic components that mirror real-world randomness. CVML's good performance on this synthetic dataset shows meaningful values for the broader time series forecasting field.

## F MORE MPTP RESULTS ON DIFFERENT FEATURE SUBSETS

Table 11 and Table 12 include the more FI-2010 results for LOB models on the MPTP task on the basic feature set and the basic+time_insensitive feature set. They support Figure 1.

## G COMPUTATIONAL RESOURCES

For all the experiments, we use a computation cluster with 1 node of 8 Nvidia A100 GPUs and 3 nodes of 8 Nvidia 2080Ti GPUs. The RAM is 1TB per node and there are 96 CPU cores per node.

Table 11: **Mid-price Trend Prediction F1 Scores (Mean&Standard Deviation) on Basic LOB data**. 13 models relevant in the literature are benchmarked to compare their F1 scores across horizons {1,2,3,5,10} on the basic LOB feature set of the FI-2010 dataset. For each horizon, the best model is bolded, and the next best model is underlined.

| | FI-2010 | | | | |
|---|---|---|---|---|---|
| Model | K=1 | K=2 | K=3 | K=5 | K=10 |
| MLP | 35.014 (4.545) | 42.242 (2.038) | 46.759 (0.918) | 46.061 (0.904) | 47.210 (2.887) |
| LSTM | 64.809 (1.377) | 57.882 (0.613) | 65.205 (0.198) | 66.898 (0.747) | 58.850 (0.926) |
| CNN1 | 27.608 (0.000) | 30.815 (0.096) | 54.783 (4.891) | 62.882 (0.828) | 63.955 (0.705) |
| CTABL | 67.353 (0.585) | 60.531 (0.213) | 66.186 (0.198) | 70.736 (0.367) | 71.244 (1.092) |
| DEEPLOB | 70.018 (1.160) | 62.357 (0.577) | 70.403 (1.010) | 75.924 (0.089) | 77.551 (0.285) |
| DAIN | 79.767 (0.050) | 70.202 (0.110) | 79.851 (0.036) | 87.041 (0.029) | 91.816 (0.075) |
| CNNLSTM | 27.620 (0.000) | 29.656 (1.210) | 34.060 (2.43) | 44.248 (10.898) | 54.872 (6.768) |
| CNN2 | 27.620 (0.000) | 27.914 (1.978) | 33.315 (1.909) | 50.086 (11.537) | 61.930 (5.501) |
| TRANSLOB | 51.020 (12.632) | 40.976 (8.987) | 50.876 (10.709) | 60.748 (1.541) | 59.715 (0.859) |
| TLONBoF | 37.549 (1.759) | 40.181 (3.560) | 41.551 (2.348) | 48.991 (1.371) | 60.702 (6.665) |
| BINCTABL | **80.985 (0.055)** | **71.168 (0.371)** | **80.734 (0.044)** | **87.553 (0.037)** | **92.074 (0.042)** |
| DEEPLOBATT | 69.435 (0.011) | 62.936 (0.015) | 59.100 (0.203) | 73.083 (0.007) | 77.028 (0.020) |
| DLA | 76.410 (0.028) | 65.966 (0.012) | 77.858 (0.006) | 85.713 (0.005) | 51.617 (0.010) |
| Mean | 55.016 | 58.290 | 65.311 | 69.695 | 72.525 |

Table 12: **Mid-price Trend Prediction F1 Scores (Mean&Standard Deviation) on Basic LOB data + time-insensitive features**. The F1 scores across horizons {1,2,3,5,10} on the FI-2010 dataset using the basic LOB + time-insensitive feature set. For each horizon, the best model is bolded, and the next best model is underlined.

| | FI-2010 | | | | |
|---|---|---|---|---|---|
| Model | K=1 | K=2 | K=3 | K=5 | K=10 |
| MLP | 45.849 (1.995) | 44.475 (1.324) | 47.855 (0.570) | 44.487 (1.209) | 49.629 (0.564) |
| LSTM | 74.261 (0.051) | 65.072 (0.223) | 72.295 (0.579) | 76.537 (1.496) | 60.141 (1.992) |
| CNN1 | 60.554 (13.194) | 61.928 (0.308) | 70.664 (0.174) | 77.906 (0.520) | 80.107 (1.099) |
| CTABL | 77.336 (0.107) | 68.265 (0.483) | 76.910 (0.217) | 82.573 (0.388) | 84.356 (0.203) |
| DEEPLOB | 79.047 (0.075) | 69.773 (0.216) | 78.797 (0.087) | 85.249 (0.090) | 88.471 (0.177) |
| DAIN | 79.935 (0.028) | **80.190 (0.102)** | 79.905 (0.072) | 87.151 (0.022) | 92.222 (0.024) |
| CNNLSTM | 38.279 (12.537) | 36.006 (1.015) | 36.181 (0.456) | 36.845 (0.342) | 76.312 (1.819) |
| CNN2 | 28.558 (0.666) | 33.529 (1.818) | 36.469 (0.541) | 39.401 (2.606) | 66.267 (7.428) |
| TRANSLOB | 69.436 (5.014) | 59.600 (3.084) | 70.478 (0.758) | 75.294 (2.101) | 71.545 (6.861) |
| TLONBoF | 43.105 (8.616) | 40.792 (1.035) | 51.367 (1.609) | 59.992 (3.051) | 65.607 (3.224) |
| BINCTABL | **81.024 (0.034)** | 71.547 (0.265) | **80.896 (0.016)** | **87.849 (0.055)** | **92.541 (0.081)** |
| DEEPLOBATT | 75.652 (0.013) | 58.878 (0.111) | 68.990 (0.103) | 67.337 (0.119) | 56.923 (0.034) |
| DLA | 77.143 (0.004) | 67.721 (0.007) | 78.241 (0.002) | 85.417 (0.002) | 58.703 (0.005) |
| Mean | 63.860 | 58.290 | 65.311 | 69.695 | 72.525 |

## H  FULL CVML RESULTS ON MPRF

Table 13 includes the full mean and standard deviation results for the time series models using CVML. Table 14 contains ablation results on CVML using its two ablated variants, CVML-abla1 and CVML-abla2. CVML-abla1 focuses exclusively on cross-variate correlations ang ignores temporal relationships by performing convolution with a kernel size of 1. CVML-abla2 uses depthwise convolution. We set the number of groups in the convolution equal to the number of input channels, which allows the model to focus exclusively on temporal correlations and ignore cross-variate correlations. The results demonstrate how these ablated versions compare to the original CVML design, highlighting the architecture's dual-correlation approach and its impact on model performance.

Table 13: **Full MPRF Results for Table 5 with Mean and Std on Basic LOB data with CVML**

| Model | Metric | K=1 | K=2 | K=3 | K=5 | K=10 |
|---|---|---|---|---|---|---|
| PatchTST | MSE | 0.653 (0.004) | 1.071 (0.016) | 1.370 (0.024) | 1.893 (0.043) | 2.646 (0.018) |
| | Corr | 0.070 (0.036) | 0.113 (0.037) | 0.165 (0.061) | 0.191 (0.045) | 0.241 (0.011) |
| | $R^2$ | 0.005 (0.005) | 0.007 (0.015) | 0.028 (0.017) | 0.033 (0.022) | 0.054 (0.007) |
| DLinear | MSE | 0.650 (0.003) | 1.042 (0.004) | 1.352 (0.011) | 1.796 (0.008) | 2.548 (0.028) |
| | Corr | 0.104 (0.021) | 0.192 (0.011) | 0.205 (0.021) | 0.291 (0.007) | 0.313 (0.024) |
| | $R^2$ | 0.010 (0.004) | 0.035 (0.004) | 0.040 (0.007) | 0.082 (0.004) | 0.089 (0.010) |
| iTransformer | MSE | 0.654 (0.002) | 1.084 (0.014) | 1.402 (0.007) | 2.002 (0.046) | 2.649 (0.014) |
| | Corr | 0.054 (0.039) | 0.070 (0.007) | 0.088 (0.014) | 0.121 (0.027) | 0.249 (0.018) |
| | $R^2$ | 0.002 (0.024) | -0.005 (0.013) | 0.005 (0.005) | -0.024 (0.024) | 0.053 (0.005) |
| TimeMixer | MSE | 0.642 (0.004) | 1.033 (0.006) | 1.329 (0.007) | 1.807 (0.029) | 2.494 (0.056) |
| | Corr | 0.160 (0.009) | 0.221 (0.009) | 0.257 (0.007) | 0.298 (0.009) | 0.353 (0.008) |
| | $R^2$ | 0.022 (0.006) | 0.043 (0.005) | 0.056 (0.005) | 0.076 (0.015) | 0.109 (0.020) |

Table 14: **MPRF Results on Ablated CVMLs**. The comparison to CVML's results in Table 5 demonstrates CVML's ability to capture cross-variate correlations and temporal correlations.

| | | CVML-abla1 (Cross-variate) | | | | | CVML-abla2 (Temporal) | | | | |
|---|---|---|---|---|---|---|---|---|---|---|---|
| Model | Metric | K=1 | K=2 | K=3 | K=5 | K=10 | K=1 | K=2 | K=3 | K=5 | K=10 |
| PatchTST | MSE | 0.6565 | 1.1168 | 1.3936 | 1.9292 | 2.7402 | 0.6531 | 1.0763 | 1.4063 | 1.9501 | 2.7983 |
| | Corr | 0.0102 | 0.0572 | 0.1080 | 0.1204 | 0.1443 | 0.0826 | 0.0813 | 0.0784 | 0.0909 | 0.0835 |
| | $R^2$ | -0.0005 | -0.0349 | 0.0107 | 0.0138 | 0.0206 | 0.0046 | 0.0026 | 0.0016 | 0.0031 | -0.0002 |
| DLinear | MSE | 0.6501 | 1.0681 | 1.3680 | 1.842 | 0.25945 | 0.6526 | 1.0727 | 1.4016 | 1.9446 | 2.7819 |
| | Corr | 0.0789 | 0.1031 | 0.1750 | 0.243 | 0.2710 | 0.0773 | 0.0810 | 0.0744 | 0.0814 | 0.0860 |
| | $R^2$ | 0.009 | 0.0102 | 0.0288 | 0.0283 | 0.0726 | 0.0055 | 0.0060 | 0.0048 | 0.0059 | 0.0057 |
| iTransformer | MSE | 0.7341 | 1.272 | 1.488 | 1.9398 | 3.1331 | 0.6967 | 1.1605 | 1.5865 | 2.2183 | 3.4300 |
| | Corr | 0.0372 | 0.061 | 0.069 | 0.1178 | 0.1000 | 0.0368 | 0.0460 | 0.0280 | 0.0646 | 0.0517 |
| | $R^2$ | -0.1188 | -0.179 | -0.056 | 0.0084 | -0.1199 | -0.0617 | -0.0754 | -0.1263 | -0.1340 | -0.2260 |
| TimeMixer | MSE | 0.637 | 1.038 | 1.329 | 1.806 | 2.508 | 0.6558 | 1.0791 | 1.4006 | 1.9031 | 2.6729 |
| | Corr | 0.180 | 0.2142 | 0.257 | 0.296 | 0.342 | 0.0837 | 0.1047 | 0.1186 | 0.2114 | 0.2617 |
| | $R^2$ | 0.030 | 0.039 | 0.057 | 0.077 | 0.104 | 0.0006 | 0.0000 | 0.0271 | 0.0271 | 0.0446 |

