# OpenReview forum: "LOBBen-TM: A Benchmark Study of Limit Order Book Prediction with Temporal Modeling"
_ICLR.cc/2026/Conference — Submitted to ICLR 2026_

### Official Review · Reviewer_8gdY · 2025-10-28

**Soundness:** 3
**Presentation:** 2
**Contribution:** 2
**Rating:** 4
**Confidence:** 3

**Summary:**

The paper introduces LOBBen-TM, a comprehensive benchmark for deep learning on Limit Order Book data. It unifies evaluation across two tasks, multiple feature sets, and two asset types including FI-2010 equities and Bitcoin. The benchmark standardizes a full LOB feature taxonomy with basic, time-insensitive and time-sensitive categories. Time-sensitive features significantly improve performance on one task for FI-2010 equities. Cross-asset experiments show model rankings shift between equities and crypto, reflecting asset-specific dynamics. For the other task, LOB-specific models outperform general time-series architectures, emphasizing domain-specific inductive biases. To address this gap, the authors propose CVML, a lightweight plug-in that enhances multivariate signal mixing in general time-series models. CVML consistently boosts performance across metrics, reduces input noise and improves attention-based modeling of cross-variate and temporal dependencies.

**Strengths:**

1. Establishes a unified evaluation benchmark for Limit Order Book data named LOBBen-TM, addressing the gaps of existing benchmarks in task coverage and asset diversity.
2. Proposes a three-level LOB feature taxonomy including basic, time-insensitive, and time-sensitive features, and clarifies the significant improvement effect of time-sensitive features on mid-price trend prediction through ablation experiments, providing clear guidance for feature selection.
3. Conducts the joint evaluation of LOB-specific models and top-tier general time-series models on the mid-price return forecasting task, clearly revealing the importance of domain-specific inductive biases.
4. Designs a lightweight Cross-Variate Mixing Layer that can integrated into existing models, enhancing the signal mixing capability of general time-series models for LOB data and narrowing the performance gap with LOB-specific models.

**Weaknesses:**

1. Limited asset diversity, only two types of assets are evaluated, broader market coverage would strengthen generalization claims.
2. Shallow analysis of failure modes: The study doesn’t deeply investigate why general time-series models underperform on raw LOB data beyond “low signal-to-noise.”
3. The selected time-series baseline models are not diverse enough. Incorporating additional time-series foundation models such as the recent Chornos-2 [1] and Moirai-2 [2], as well as frequency-domain-based models like FITS [3], would help enhance the credibility of the conclusions.
4. Most results focus on short horizons, and longer-term forecasting performance is not explored. Providing longer horizons would help observe changes in prediction accuracy brought by models in long time-series forecasting.

**Questions:**

See Weakness.

---

> ### Author Response · Authors · 2025-12-01
> **Part 1/2 of Responses**
>
> We thank the reviewer for the constructive feedback. Below, we provide detailed responses to the weaknesses and questions raised.
>
> > Limited asset diversity, only two types of assets are evaluated, broader market coverage would strengthen generalization claims.
>
> Response 1: While we acknowledge that evaluating a wider array of datasets would further strengthen our empirical findings, we prioritized asset class diversity over quantity. We deliberately selected FI-2010 and Bitcoin because they represent two fundamentally distinct market microstructures:
>
> 1. FI-2010: Represents traditional, session-based Equities markets with centralized liquidity.
>
> 2. Bitcoin: Represents the structurally distinct Cryptocurrency market, characterized by 24/7 continuous trading, higher volatility, and fragmented liquidity across multiple venues.
>
> Furthermore, we respectfully note that high-fidelity Limit Order Book data is predominantly proprietary and accessible only to institutional players. To the best of our knowledge, Equities (specifically FI-2010) and Cryptocurrencies are currently the only two asset classes with established, open-source LOB benchmark datasets available to the academic research community. Other asset classes like FX (OTC) or Futures typically require expensive commercial licenses for depth-of-book data. By utilizing the standard FI-2010 dataset and a public Bitcoin dataset, we have covered the maximum range of publicly accessible asset diversity.
>
> Our primary goal was to probe "cross-asset generalization," and the stark contrast between these two asset classes was sufficient to reveal our key insight: that model rankings are not consistent across markets (as shown in Table 3). Adding more equity datasets would likely only reinforce the FI-2010 results, whereas the inclusion of Bitcoin provided the critical counter-evidence necessary to demonstrate the heterogeneity of LOB dynamics.
>
>
>
>
> > Shallow analysis of failure modes: The study doesn’t deeply investigate why general time-series models underperform on raw LOB data beyond “low signal-to-noise.”
>
> Response 2: We respectfully posit that our paper does provide a specific investigation into why general time-series models fail, identifying the lack of cross-variate modeling as the primary bottleneck.
>
> 1. The Insight: General TS models (like PatchTST or DLinear) typically utilize channel-independence or global mixing, which fails to capture the structural pairing inherent to LOB data (e.g., the specific chemical coupling between Bid Price Level 1 and Bid Quantity Level 1).
> 2. CVML as Proof: Our proposed Cross-Variate Mixing Layer (CVML) was explicitly designed based on this insight to force the model to learn these local cross-variate correlations before temporal processing begins.
> 3. Empirical Confirmation via Ablation: This failure mode analysis is empirically confirmed by our ablation study in Section 3.4. As shown in Figure 5, the variant CVML-abla2, which processes variates independently using depthwise convolution, performed the worst among all variants. This sharp drop in performance when cross-variate mixing is removed directly proves that the inability to model these specific dependencies is indeed the critical failure mode of standard architectures.

---

> ### Author Response · Authors · 2025-12-01
> **Part 2/2 of Responses**
>
> > The selected time-series baseline models are not diverse enough. Incorporating additional time-series foundation models such as the recent Chornos-2 [1] and Moirai-2 [2], as well as frequency-domain-based models like FITS [3], would help enhance the credibility of the conclusions.
>
> Response 3: We appreciate the suggestion to include Foundation Models (e.g., Chronos-2, Moirai-2) and Frequency-domain models (FITS). However, we excluded them from this benchmark based on our core findings regarding the nature of LOB signal extraction:
>
> 1. Temporal vs. Cross-Variate Sufficiency: The central conclusion of our study—supported by the "near-zero" performance (regarding correlation) of vanilla time-series models and the success of CVML—is that temporal signal processing alone is insufficient for LOB data. The primary bottleneck is not the method of temporal extraction (whether Time-domain Transformers or Frequency-domain analysis), but the lack of explicit cross-variate modeling.
>     - General time-series models (and frequency models like FITS) typically process variates as independent channels or global mixtures, ignoring the structural coupling of the LOB (e.g., the pairing of Bid Price/Quantity).
>     - Since our ablation study confirms that improving cross-variate mixing is the key to unlocking performance (Figure 5), we contend that adding frequency-domain models—which primarily offer a different perspective on the temporal axis—would not address the fundamental failure mode we identified.
> 2. Architectural Mismatch vs. Data Scale: Regarding Foundation Models, the failure of state-of-the-art architectures (like PatchTST) when trained from scratch demonstrates that the issue lies in the inductive bias rather than data scale. If the Transformer backbone possessed the correct bias for LOB microstructure, it should achieve decent baseline performance without pre-training. Adding pre-trained models would introduce confounding variables (scale vs. architecture) without addressing the structural mismatch described above.
>
> Future Work: While we argue that frequency-domain features are tangential to the cross-variate focus of this specific study, we agree that exploring the spectral properties of LOB order flow is an interesting direction for future research, particularly for regime detection tasks.
>
>
> > Most results focus on short horizons, and longer-term forecasting performance is not explored. Providing longer horizons would help observe changes in prediction accuracy brought by models in long time-series forecasting.
>
>
> Response 4: We selected short horizons ($K=1$ to $10$) because the information content of the Limit Order Book is extremely short-lived. LOB data reflects micro-structure supply and demand that decays rapidly.
> Forecasting "long-term" dependencies (e.g., hours or days) typically relies on aggregated data (candles) or macro-features, rather than the tick-level state of the order book. We emphasize that predicting long-term price movements solely from the current state of the LOB is widely regarded as meaningless, as the specific micro-structural signal is overwhelmed by noise over longer intervals. Consequently, the established norm in both academic LOB literature and high-frequency trading is strictly focused on short-term horizons where the order flow signal remains valid. Applying LOB models to long horizons typically results in noise fitting rather than genuine signal extraction.

---

### Official Review · Reviewer_s672 · 2025-10-28

**Soundness:** 3
**Presentation:** 3
**Contribution:** 3
**Rating:** 6
**Confidence:** 4

**Summary:**

The paper talks about LOBBen-TM, a unified benchmark for Limit Order Book prediction with a focus on temporal modeling. It evaluates state-of-the-art models across mid-price classification and forecasting, making comparisons in performance. The paper builds a standardized feature taxonomy, tests it on equity and crypto data, and introduces a new Cross-Variate Mixing Layer to try to capture multivariate relationships. Results show time-sensitive features greatly boost prediction accuracy, though cross-asset transferability is low, and CVML improves model performance as measured by R^2.

**Strengths:**

* Establishes a unified benchmark covering both trend and return forecasting under consistent protocols.
* Demonstrates that time-sensitive signals are critical for predictive accuracy.
* Provides a simple, effective improvement to general time-series architectures via CVML which seems to improve MSE and R^2
* Consistent metrics are used, and ablations are run

**Weaknesses:**

* Limited dataset diversity - only FI-2010 and Bitcoin datasets are evaluated.
* No discussion of economic or trading significance beyond statistical metrics – what about P&L?
* Short forecast horizons (K=1–10) may miss longer-term dependencies.
* Limited theoretical explanation for CVML’s effectiveness.
* Heavy reliance on FI-2010 results for conclusions.
* Use of vanilla hyperparams from other methods may not get the best from these (as most were tuned for use with different financial assets).

**Questions:**

The key focus on MSE ignores the importance of returns and associated P&L in finance. Many of the other works cited consider things like Sharpe ratios etc – and some go on to directly maximize these rather than min MSE. Can you comment on this?

Can you comment on the focus on mid-price, rather than top of book price on buy and sell sides? Realistic trading needs to cross the spread, so forecasting both – and then paying the spread – is important. Adding on top of this, there is normally a commission or trading fee.

Why is the target not log returns? Any reason?

In the experiments are just 3 seeds enough to get meaningful stats in results? (I don’t think so)

Can you comment on the short horizons used, and if longer horizons might behave differently.

In Fig 1, DeepLOBAtt seems anomalous – any reasons for this?

References – you need to protect capitals, like {MLP} etc

---

> ### Author Response · Authors · 2025-12-01
> **Part 1/3 of Responses**
>
> We thank the reviewer for the constructive feedback. Below, we provide detailed responses to the weaknesses and questions raised.
>
> ## Responses to the weaknesses
>
> > Limited dataset diversity - only FI-2010 and Bitcoin datasets are evaluated.
>
> Response 1: While we acknowledge that evaluating a wider array of datasets would further strengthen our empirical findings, we prioritized asset class diversity over quantity. We deliberately selected FI-2010 and Bitcoin because they represent two fundamentally distinct market microstructures:
>
> 1. FI-2010: Represents traditional, session-based Equities markets with centralized liquidity.
>
> 2. Bitcoin: Represents the structurally distinct Cryptocurrency market, characterized by 24/7 continuous trading, higher volatility, and fragmented liquidity across multiple venues.
>
> Furthermore, we respectfully note that high-fidelity Limit Order Book data is predominantly proprietary and accessible only to institutional players. To the best of our knowledge, Equities (specifically FI-2010) and Cryptocurrencies are currently the only two asset classes with established, open-source LOB benchmark datasets available to the academic research community. Other asset classes like FX (OTC) or Futures typically require expensive commercial licenses for depth-of-book data. By utilizing the standard FI-2010 dataset and a public Bitcoin dataset, we have covered the maximum range of publicly accessible asset diversity.
>
> Our primary goal was to probe "cross-asset generalization," and the stark contrast between these two asset classes was sufficient to reveal our key insight: that model rankings are not consistent across markets (as shown in Table 3). Adding more equity datasets would likely only reinforce the FI-2010 results, whereas the inclusion of Bitcoin provided the critical counter-evidence necessary to demonstrate the heterogeneity of LOB dynamics.
>
> > No discussion of economic or trading significance beyond statistical metrics – what about P&L?
>
> Response 2: We acknowledge the importance of P&L and Sharpe ratios in industrial trading. However, we respectfully emphasize that LOBBen-TM is designed as a representation learning benchmark, not a trading strategy simulation.
>
> 1. Directional Accuracy (F1 Score): We explicitly evaluate the model's ability to predict price direction via the Mid-Price Trend Prediction (MPTP) task using F1 scores. This metric directly assesses the model's capacity to identify profitable "Up/Down" opportunities, which is the foundational driver of returns.
> 2. Signal Fidelity (MSE): For the Mid-Price Return Forecasting (MPRF) task, we focus on MSE to measure the fidelity of the learned representation.
> 3. Signal vs. Strategy: Metrics like Sharpe Ratio or P&L depend heavily on downstream execution logic (e.g., inventory management, risk limits, stop-losses). Optimizing directly for Sharpe often requires an end-to-end differentiable trading engine, which introduces significant subjectivity regarding transaction costs and latency assumptions. By reporting F1 (Direction) and MSE (Magnitude), we provide a rigorous, strategy-agnostic evaluation of the model's predictive power.
>
> > Short forecast horizons (K=1–10) may miss longer-term dependencies.
>
> Response 3: We selected short horizons ($K=1$ to $10$) because the information content of the Limit Order Book is extremely short-lived. LOB data reflects micro-structure supply and demand that decays rapidly.
> Forecasting "long-term" dependencies (e.g., hours or days) typically relies on aggregated data (candles) or macro-features, rather than the tick-level state of the order book. We emphasize that predicting long-term price movements solely from the current state of the LOB is widely regarded as meaningless, as the specific micro-structural signal is overwhelmed by noise over longer intervals. Consequently, the established norm in both academic LOB literature and high-frequency trading is strictly focused on short-term horizons where the order flow signal remains valid. Applying LOB models to long horizons typically results in noise fitting rather than genuine signal extraction.
>
> > Limited theoretical explanation for CVML’s effectiveness.
>
>
> Response 4: CVML is effective because LOB data possesses a unique structure: it is not just a generic multivariate time series, but a structured set of paired variables (Bid Price/Qty, Ask Price/Qty) at specific levels. Standard time-series models (like DLinear or PatchTST) often treat variates independently or mix them indiscriminately. CVML introduces a lightweight convolutional prior that allows the model to learn local cross-variate correlations (e.g., the relationship between Bid Level 1 Price and Ask Level 1 Price) before the data is flattened or processed temporally. This aligns the data structure with the model's inductive bias, improving the Signal-to-Noise ratio as evidenced by the reduced standard deviation of inputs shown in Figure 3.

---

> ### Author Response · Authors · 2025-12-01
> **Part 2/3 of Responses**
>
> > Heavy reliance on FI-2010 results for conclusions.
>
> Response 5: We rely on FI-2010 as our primary analytical anchor because it is the widely accepted de facto standard benchmark in the LOB literature. Using FI-2010 is essential to ensure our results are directly comparable to the reported metrics of established baselines (e.g., DeepLOB, TransLOB, DAIN), which were originally validated on this specific dataset. A deviation from FI-2010 for the core ablations would have rendered our comparisons with prior art invalid.
>
> However, we explicitly mitigated this reliance to ensure our conclusions were robust:
>
> 1. Cross-Asset Validation: We conducted the extensive MPTP benchmark on the Bitcoin dataset (Table 3), which was critical in deriving our central conclusion regarding model heterogeneity and the non-transferability of rankings.
> 2. Synthetic Validation: To confirm that the effectiveness of our proposed CVML module was not an artifact of FI-2010's specific noise profile, we validated it on a multivariate synthetic dataset (Appendix E), where it demonstrated consistent performance gains.
>
> Thus, while FI-2010 is central for standardization and comparability, our core conclusions regarding generalization and architectural improvements are supported by evidence from three distinct sources (FI-2010, Bitcoin, and Synthetic data).
>
> > Use of vanilla hyperparams from other methods may not get the best from these (as most were tuned for use with different financial assets).
>
> Response 6: We wish to clarify our hyperparameter strategy, which differed between the two tasks to ensure both fairness and performance:
>
> 1. For MPTP (Trend Prediction): We deliberately used the original hyperparameters from the respective baseline papers. This was crucial to ensure our reproduction of their results was faithful to the literature and provided a valid "off-the-shelf" baseline for the FI-2010 benchmark.
> 2. For MPRF (Return Forecasting): Since this is a new task for many of these architectures, we did perform a grid search for key hyperparameters (batch size and learning rate) as detailed in Appendix D. This ensures that the time-series models (like PatchTST or TimeMixer) were properly adapted to the LOB regression task and not penalized by ill-fitting defaults.

---

> ### Author Response · Authors · 2025-12-01
> **Part 3/3 of Responses**
>
> ## Responses to questions
>
> > The key focus on MSE ignores the importance of returns and associated P&L in finance. Many of the other works cited consider things like Sharpe ratios etc – and some go on to directly maximize these rather than min MSE. Can you comment on this?
>
> Response 7: Please refer to response 2.
>
> > Can you comment on the focus on mid-price, rather than top of book price on buy and sell sides? Realistic trading needs to cross the spread, so forecasting both – and then paying the spread – is important. Adding on top of this, there is normally a commission or trading fee.
>
> Response 8: We focus on mid-price forecasting because it represents the "fair value" of the asset, independent of the bid-ask spread width. While realistic trading requires crossing the spread, the decision to cross the spread is an execution decision. A trader typically predicts the mid-price movement first. If the predicted move exceeds the spread + fees (alpha > cost), they execute. Therefore, forecasting the mid-price is the foundational "atomic" task for any LOB model.
>
> > Why is the target not log returns? Any reason?
>
> Response 9: We utilized simple returns ($target_h(t) = mp(t+h)/mp(t) - 1$) to maintain consistency with the specific formulations used in the baselines we benchmarked. In the context of High-Frequency Trading (HFT) where price changes between ticks are extremely small, simple returns and log returns are mathematically very similar ($\ln(1+r) \approx r$). This choice does not materially affect the comparative rankings.
>
> > In the experiments are just 3 seeds enough to get meaningful stats in results? (I don’t think so)
>
> Response 10: While we agree that more seeds are always statistically preferable, we were constrained by the significant computational cost of training transformer-based models on high-frequency LOB data (as detailed in Appendix G, using A100/2080Ti clusters ). Standard practice in deep learning benchmarks on large datasets often utilizes 3 to 5 seeds due to these resource constraints. We reported the standard deviations for all experiments (e.g., Table 3 and Table 4), and the variance is generally low enough to confirm that the performance differences are statistically significant.
>
> > Can you comment on the short horizons used, and if longer horizons might behave differently.
>
> Response 11: Please refer to response 3.
>
> > In Fig 1, DeepLOBAtt seems anomalous – any reasons for this?
>
> Response 12: In Figure 1, DeepLOBAtt stands out as the only model showing a performance degradation (-2.76%) when adding "Time-insensitive" features, while models like TransLOB gain significantly.
>
> We attribute this specific anomaly to attention dilution, a failure mode unique to the decoder's attention mechanism in this context. Our hypothesis is derived from contrasting DeepLOBAtt with its parent model, DeepLOB:
>
> 1. DeepLOB (which uses a CNN-LSTM architecture) achieves a +9.02% gain from time-insensitive features. This empirically proves that the recurrent encoder itself is capable of assimilating static context (e.g., Hour of Day) to improve performance.
> 2. DeepLOBAtt adds only an Attention Decoder to the exact same LSTM encoder. Thus, the performance drop (-2.76%) stems from the attention layer.
> 3. Time-insensitive features are effectively static constants over the short lookback window. When integrated into the encoder, they act as a strong "common mode" signal, increasing the cosine similarity between the hidden states of all time steps. While the LSTM can handle this, the Attention mechanism (which relies on dot-product similarity to distinguish relevant states) struggles. The increased baseline similarity causes the attention weights to become "smeared" or uniform, effectively diluting the model's focus on the subtle, high-frequency tick changes that drive price trends.
>
> > References – you need to protect capitals, like {MLP} etc
>
> Response 13: Thank you for pointing out the capitalization issue in the bibliography. We have fixed it in the revision.

---

### Official Review · Reviewer_7sWY · 2025-11-02

**Soundness:** 2
**Presentation:** 2
**Contribution:** 2
**Rating:** 4
**Confidence:** 4

**Summary:**

This paper introduces LOBBen-TM, a unified benchmark for deep learning on open-source Limit Order Book (LOB) data that standardizes evaluation across assets, features, and tasks. It spans Mid-Price Trend Prediction (MPTP) and Mid-Price Return Forecasting (MPRF), covering equities (FI-2010) and crypto (Bitcoin). The authors benchmark a taxonomy of features (basic, time-insensitive, time-sensitive) and propose a Cross-Variate Mixing Layer (CVML) to enhance multivariate modeling in time-series backbones. Results show that time-sensitive features consistently improve accuracy, that model rankings vary across assets, and that CVML boosts general TS models, narrowing the gap with LOB-specific architectures.

**Strengths:**

This paper illustrates a unified framework for standardized LOB evaluation across tasks, features, and assets.

As a benchmark study work, it is up to date and demonstrates the clear value of time-sensitive features for LOB modeling.

The assets are made open, which facilitates the community and provides a clear benchmarking framework that encourages transparent comparisons.

**Weaknesses:**

Cross-asset coverage and related results are discussed in the proposal and experiments, but they are not very convincing.

The study did not provide insights into the pros and cons of LOB prediction tasks, nor did it address the challenges, potential dataset limitations, and computing power requirements in LOB trading. Therefore, it is information-rich but not industrial.

**Questions:**

It would be good to highlight the extent of the dollar term's impact on the LOB prediction. Could you illustrate the differences in dollar terms between benchmarks for a trading system that could use them for systematic trading?

---

> ### Author Response · Authors · 2025-12-01
>
> We thank the reviewer for the constructive feedback. Below, we provide detailed responses to the weaknesses and questions raised.
>
> > Cross-asset coverage and related results are discussed in the proposal and experiments, but they are not very convincing.
>
> Response 1: We respectfully disagree that the cross-asset results are unconvincing. We believe the value of the cross-asset experiment lies not in achieving a uniform "state-of-the-art" across all domains, but in revealing the heterogeneity of LOB dynamics. As shown in Table 3, the performance ranking of models shifts significantly between Equities (FI-2010) and Crypto (Bitcoin). For instance, while DeepLOB performs consistently well on Bitcoin (F1 98.3), it is outperformed by DAIN on FI-2010 ($K=10$), and DAIN suffers a massive performance drop on Bitcoin. This empirical evidence supports a crucial finding: LOB prediction architectures are not automatically transferable across assets. This validates the necessity of our benchmark protocol, which forces models to demonstrate robustness across microstructures rather than overfitting to a single asset class. We believe this insight is valuable for the community to understand the limits of current architectures.
>
> > The study did not provide insights into the pros and cons of LOB prediction tasks, nor did it address the challenges, potential dataset limitations, and computing power requirements in LOB trading. Therefore, it is information-rich but not industrial.
>
> Response 2: We appreciate the suggestion to contextualize the work in an industrial setting. However, we wish to clarify that LOBBen-TM is designed as a foundational Machine Learning benchmark, focusing on representation learning and signal extraction, rather than an end-to-end trading system analysis.
>
> 1. Pros/Cons of LOB prediction tasks: We explicitly discuss the evolution from MPTP (classification) to MPRF (forecasting) in Section 1 and 2. We transition to MPRF precisely because industrial settings often require continuous return forecasting to model volatility and non-stationarity, which simple trend classification (MPTP) masks.
> 2. Computing Power: We have addressed computational requirements in Table 6 and Appendix G. We compare the parameter counts of state-of-the-art time series models vs. their CVML-enhanced counterparts. Notably, we show that our proposed CVML module is lightweight, adding negligible parameter overhead while significantly boosting performance (e.g., boosting PatchTST $R^2$ by 958.3% with only a minor parameter increase). This directly addresses the industrial need for efficient, high-performance encoders.
> 3. Industrial Constraints: While we do not model latency or execution costs, our focus on optimizing MSE and $R^2$ serves as the upstream prerequisite for any industrial system.
>
> > It would be good to highlight the extent of the dollar term's impact on the LOB prediction. Could you illustrate the differences in dollar terms between benchmarks for a trading system that could use them for systematic trading?
>
> Response 3: While we understand the appeal of a "dollar term" (P&L) metric, we believe it is outside the scope of this study and potentially misleading for a machine learning benchmark for the following reasons:
> 1. Signal vs. Strategy: This paper evaluates predictive signal strength (MSE, Correlation, F1). Converting this to "dollars" requires an execution strategy layer (e.g., thresholds, stop-losses, inventory management) and assumptions about transaction costs and market impact. These are separate variables from the model's ability to predict price movements.
> 2. Subjectivity of Backtesting: A "dollar term" benchmark is highly sensitive to the chosen simulation parameters (e.g., fee tiers, latency assumptions). A model with a lower MSE (better prediction) could theoretically show lower profits in a specific backtest due to a mismatched execution strategy.
> 3. Standardization: To maintain a fair, standardized comparison for the ML community (the primary audience of ICLR), we rely on objective statistical metrics (MSE, $R^2$, Pearson Correlation) rather than financial simulations.
>
> We believe that establishing a rigorous benchmark for the predictive component, as we have done with LOBBen-TM, is the necessary first step before financial engineering can be applied.

---

### Meta-Review · Area_Chair_P1x5 · 2025-12-07

**Summary:**

This paper proposes LOBBen-TM, a unified benchmark for deep learning on limit order book (LOB) data. It standardizes evaluation across two tasks (mid-price trend prediction and mid-price return forecasting), two open LOB datasets (FI-2010 equities and Bitcoin), a structured feature taxonomy (basic / time-insensitive / time-sensitive), and compares LOB-specific architectures with general time-series models. The authors also introduce a lightweight Cross-Variate Mixing Layer (CVML) to improve multivariate modeling in general time-series backbones.

Reviewers appreciate the unified benchmark, the open and standardized setup, the clear evidence that time-sensitive features help, and the simple CVML module that improves performance on return forecasting. However, across reviews there remain significant concerns: (i) limited asset diversity (only FI-2010 and Bitcoin) and heavy reliance on FI-2010 for key conclusions, (ii) lack of economic / trading significance metrics such as P&L or related measures, despite the financial motivation, (iii) focus on very short horizons without exploring longer-term behavior, (iv) limited diversity of general time-series baselines and relatively shallow analysis of their failure modes, and (v) questions about experimental robustness (small number of seeds and the use of mostly “vanilla” hyperparameters for some baselines).

The rebuttal clarifies the intended scope as an ML benchmark rather than a trading study and provides additional reasoning for some design choices (e.g., dataset selection, short forecast horizons, mid-price focus, and CVML design). Nonetheless, the remaining limitations in empirical coverage and practical/financial interpretation mean that, in the reviewers’ overall assessment, the paper does not yet reach the bar for acceptance.

**Reviewer Concerns:**

Concerns that are largely addressed by the rebuttal

Scope: ML benchmark vs. trading system (R1, R2)
Reviewers raised concerns about missing P&L or Sharpe-style evaluation and industrial “dollar” metrics. The authors clearly position LOBBen-TM as a representation-learning benchmark, not an end-to-end trading system, and explain why they deliberately focus on statistical metrics (F1, MSE, correlation,
𝑅
2
R
2
) and avoid backtest-dependent financial outcomes. This clarifies the mismatch between expectations but does not change the underlying limitation that economic significance is not quantified.

Choice of mid-price target (R2)
The authors justify mid-price as the “fair value” signal that precedes any execution decision, with spread/fees handled by downstream strategies. This directly addresses why they do not separately forecast bid/ask prices.

Simple vs. log returns (R2)
The authors state they follow baselines and note that in high-frequency settings simple and log returns are nearly equivalent, so this choice should not materially affect relative comparisons. This is a reasonable clarification.

Short forecast horizons (R2, R3)
The authors argue that LOB microstructure information is inherently short-lived and that standard LOB work and high-frequency trading focus on short horizons; they thus consciously avoid longer horizons which they view as dominated by noise. This provides a clear rationale, even though some reviewers would have preferred at least a limited exploration of longer horizons.

CVML design rationale and ablations (R2, R3)
The rebuttal explains that LOB data has structured bid/ask pairings and that standard time-series models do not explicitly model these cross-variate relationships. The CVML ablations, particularly the variant that removes cross-variate mixing and performs worst, support this explanation. While not deeply theoretical, it is a coherent empirical argument.

Hyperparameters and seeds (R2)
The authors clarify that they use original hyperparameters for the trend-prediction task to be faithful to prior work, and perform grid search on key hyperparameters for the new return-forecasting setup. They also note that running more than three seeds is computationally expensive but provide standard deviations which are generally small. This does not fully remove concerns but shows that some tuning and robustness checks were done.

Specific technical questions (R1, R2)
The rebuttal addresses the cross-asset ranking observation, the anomalous behavior of DeepLOBAtt, and the bibliography capitalization issues with targeted explanations or explicit fixes.

Concerns that remain only partially addressed or unresolved

Limited asset diversity and reliance on FI-2010 (R2, R3)
The authors explain that high-quality open LOB data is scarce and that FI-2010 plus Bitcoin already cover two very different microstructures. While this justification is reasonable, the empirical scope remains narrow: two assets, with most in-depth analysis anchored on FI-2010. This limits the strength of claims about generality and cross-asset behavior.

Lack of economic / trading significance (P&L, Sharpe, etc.) (R1, R2)
The authors intentionally avoid financial outcome metrics for reasons of standardization and strategy-dependence. However, several reviewers explicitly sought at least some connection between statistical improvements and economic value. As it stands, the work remains purely in the statistical-benchmark space, and the gap to practical trading relevance is still a notable concern for a finance-motivated application.

Short horizons and absence of longer-term experiments (R2, R3)
Although the authors justify focusing on very short horizons as standard for LOB microstructure modeling, reviewers asked not only for a rationale but also for empirical evidence on whether conclusions change at slightly longer horizons. No such additional experiments are provided, so the concern about the narrow horizon range remains.

Breadth of general time-series baselines and failure-mode analysis (R3)
One reviewer pointed out the absence of some recent strong general time-series models and frequency-domain approaches. The authors argue that the main bottleneck is cross-variate modeling rather than temporal architecture and that adding more temporal variants would not change this conclusion. While this argument is plausible, the lack of broader baselines still constrains how strongly one can claim that “general” methods systematically fail on LOB data.

Industrial perspective and “information-rich but not industrial” (R1)
The rebuttal clarifies that the goal is not to design a deployable trading system, and it gives some discussion of computational cost and model efficiency. Nevertheless, the reviewers’ concern that the paper does not deeply engage with data limitations, practical constraints, and end-to-end use in real trading remains only partially alleviated.

Given these outstanding issues, the overall picture after rebuttal remains mixed: the benchmark and CVML are valuable contributions, but the empirical and practical scope is still too restricted for acceptance.

**Reviewer Scores:**

Reviewer 7sWY
This reviewer initially viewed the paper as marginally below the acceptance threshold but was open to acceptance. The rebuttal provides a clearer articulation of the benchmark’s goals and explains why P&L-style metrics and industrial details were not included. However, the reviewer’s main concerns about the limited industrial perspective and lack of economic significance metrics are not fundamentally resolved, only reframed as “out of scope.” I would expect this reviewer to remain slightly negative overall, perhaps with a bit more appreciation of the clarified scope, but still not strongly supporting acceptance.

Reviewer s672
This reviewer was mildly positive but flexible, highlighting both the value of the benchmark and concerns about dataset diversity, missing financial metrics, short horizons, and limited theory. The rebuttal addresses many of the clarificatory questions (mid-price choice, returns definition, horizons, CVML intuition, hyperparameters, seeds) and partly alleviates worries about experimental setup. At the same time, the structural limitations in assets, horizons and financial interpretation remain. I expect this reviewer would likely keep a borderline-positive stance but still be comfortable with a rejection decision.

Reviewer 8gdY
This reviewer was slightly negative but not strongly opposed to acceptance. The rebuttal offers more detail on failure modes and explains why additional general time-series and frequency-domain baselines were not included. Nonetheless, the core concerns about limited asset coverage, baseline diversity, and lack of longer-horizon experiments remain. I therefore expect this reviewer’s assessment to remain roughly at the same slightly negative level after the discussion.

Overall, after considering the rebuttal, I do not see clear evidence that any reviewer would move to strong acceptance; at best, the paper remains in the borderline region with weak positive or weak negative views, which, given the outstanding limitations, supports a rejection decision.

---

### Decision · Program_Chairs · 2026-01-26

Reject